# Broadly neutralizing antibodies target a haemagglutinin anchor epitope

Jenna J. Guthmiller[1,18 ✉], Julianna Han[2,18], Henry A. Utset[1], Lei Li[1], Linda Yu-Ling Lan[3], Carole Henry[1,16], Christopher T. Stamper[3], Meagan McMahon[4], George O'Dell[4], Monica L. Fernández-Quintero[5], Alec W. Freyn[4,16], Fatima Amanat[4], Olivia Stovicek[1], Lauren Gentles[6,7], Sara T. Richey[2], Alba Torrents de la Peña[2], Victoria Rosado[4], Haley L. Dugan[3], Nai-Ying Zheng[1], Micah E. Tepora[1], Dalia J. Bitar[1], Siriruk Changrob[1], Shirin Strohmeier[4], Min Huang[1], Adolfo García-Sastre[4,8,9,10,11], Klaus R. Liedl[5], Jesse D. Bloom[6,7,12,13], Raffael Nachbagauer[4,16], Peter Palese[4,8], Florian Krammer[4], Lynda Coughlan[14,15], Andrew B. Ward[2 ✉] & Patrick C. Wilson[1,3,17 ✉]

Broadly neutralizing antibodies that target epitopes of haemagglutinin on the influenza virus have the potential to provide near universal protection against influenza virus infection[1]. However, viral mutants that escape broadly neutralizing antibodies have been reported[2,3]. The identification of broadly neutralizing antibody classes that can neutralize viral escape mutants is critical for universal influenza virus vaccine design. Here we report a distinct class of broadly neutralizing antibodies that target a discrete membrane-proximal anchor epitope of the haemagglutinin stalk domain. Anchor epitope-targeting antibodies are broadly neutralizing across H1 viruses and can cross-react with H2 and H5 viruses that are a pandemic threat. Antibodies that target this anchor epitope utilize a highly restricted repertoire, which encodes two public binding motifs that make extensive contacts with conserved residues in the fusion peptide. Moreover, anchor epitope-targeting B cells are common in the human memory B cell repertoire and were recalled in humans by an oil-in-water adjuvanted chimeric haemagglutinin vaccine[4,5], which is a potential universal influenza virus vaccine. To maximize protection against seasonal and pandemic influenza viruses, vaccines should aim to boost this previously untapped source of broadly neutralizing antibodies that are widespread in the human memory B cell pool.

Antibodies to the major surface glycoprotein haemagglutinin (HA) are critical for providing protection against influenza virus infection[6,7]. However, most HA-binding antibodies target variable epitopes of the HA head domain, which provide limited protection against antigenically similar influenza virus strains[3]. Vaccine formulations that preferentially induce antibodies to conserved epitopes of the HA head and stalk domains could provide broad and potent protection against a wide array of influenza viruses. Several leading universal influenza virus candidates are designed to induce antibodies specifically to the stalk domain, but the spectrum of distinct epitopes on the stalk targeted by the human B cell repertoire remains to be determined. By analysing the specificities of B cells targeting the H1 stalk through the generation of monoclonal antibodies (mAbs), our study reveals a new class of broadly

neutralizing antibodies (bnAbs) to an underappreciated epitope where HA anchors itself into the viral membrane. Next-generation vaccine platforms should take advantage of this finding to elicit antibodies to the conserved anchor epitope.

## Discovery of anchor epitope-binding mAbs

To investigate the specificities of HA-specific antibodies, we generated 358 mAbs from plasmablasts and HA+ memory B cells (MBCs) isolated from volunteers who were vaccinated against or naturally infected with seasonal influenza viruses or were participants in a phase I clinical trial of a chimeric HA (cHA) vaccine[4,5]. Of all mAbs tested, nearly half targeted the HA stalk domain, 21% of which targeted the well-characterized

[1]Department of Medicine, Section of Rheumatology, University of Chicago, Chicago, IL, USA. [2]Department of Integrative Structural and Computational Biology, The Scripps Research Institute, La Jolla, CA, USA. [3]Committee on Immunology, University of Chicago, Chicago, IL, USA. [4]Department of Microbiology, Icahn School of Medicine at Mount Sinai, New York, NY, USA. [5]Center for Molecular Biosciences Innsbruck, Department of General, Inorganic and Theoretical Chemistry, University of Innsbruck, Innsbruck, Austria. [6]Basic Sciences Division, Fred Hutchinson Cancer Research Center, Seattle, WA, USA. [7]Department of Microbiology, University of Washington, Seattle, WA, USA. [8]Department of Medicine, Division of Infectious Diseases, Icahn School of Medicine at Mount Sinai, New York, NY, USA. [9]Global Health and Emerging Pathogens Institute, Icahn School of Medicine at Mount Sinai, New York, NY, USA. [10]The Tisch Cancer Center, Icahn School of Medicine at Mount Sinai, New York, NY, USA. [11]Department of Pathology, Molecular and Cell-Based Medicine, Icahn School of Medicine at Mount Sinai, New York, NY, USA. [12]Department of Genome Sciences, University of Washington, Seattle, WA, USA. [13]Howard Hughes Medical Institute, Fred Hutchinson Cancer Research Center, Seattle, WA, USA. [14]Department of Microbiology and Immunology, University of Maryland School of Medicine, Baltimore, MD, USA. [15]Center for Vaccine Development and Global Health (CVD), University of Maryland School of Medicine, Baltimore, MD, USA. [16]Present address: Moderna Inc., Cambridge, MA, USA. [17]Present address: Drukier Institute for Children's Health, Department of Pediatrics, Weill Cornell Medicine, New York, NY, USA. [18]These authors contributed equally: Jenna J. Guthmiller, Julianna Han. ✉e-mail: jguthmiller@uchicago.edu; andrew@scripps.edu; pcw4001@med.cornell.edu

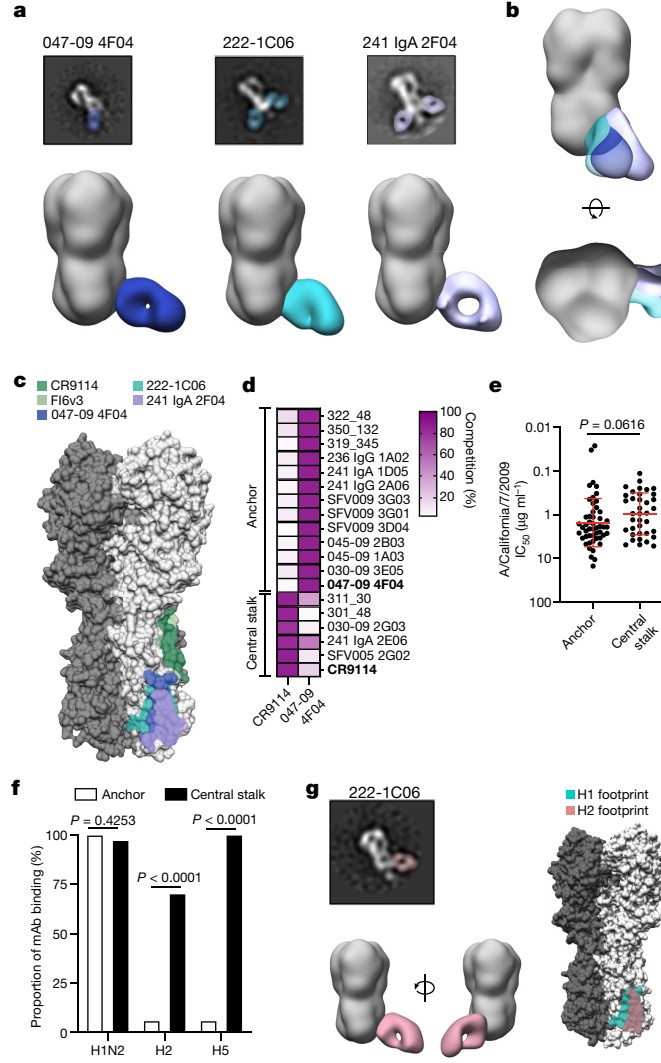

three mAbs bound an epitope near the anchor of the HA stalk and were oriented at an upward angle towards the epitope (Fig. 1a, b, Extended Data Fig. 1c), suggesting that this epitope may be partially obstructed by the lipid membrane and may only be exposed for antibody binding as the HA trimers flex on the membrane. FISW84, a recently identified stalk-binding mAb[11], targets an epitope that overlaps with the three identified anchor-binding mAbs (Extended Data Fig. 1d), suggesting that the anchor epitope is a common stalk epitope. Moreover, a proximal epitope was previously identified on group 2 viruses that is targeted by mAbs CR8020 (ref. [12]) and CR8043 (ref. [13]). Despite some overlap, the epitope targeted by the group 2 mAbs was considerably farther up and to the right on the HA stalk relative to the anchor epitope, and mAbs CR8020 and CR8043 targeted the stalk from above (Extended Data Fig. 1e, f), at an angle and positioning more akin to antibodies targeting the CS epitope. mAbs binding the CS epitope (CR9114 (ref. [10]) and FI6v3 (ref. [14])) did not have overlapping footprints or compete for HA binding with the anchor-binding 047-09 4F04 mAb in an HA competition assay (Fig. 1c, d). In total, we identified 50 distinct mAbs that competed for binding to the anchor epitope from a total of 21 individuals (Fig. 1d, Extended Data Tables 1, 2). Of these, 34 anchor-binding mAbs from 15 donors were isolated from the cHA vaccine trial (Extended Data Tables 1, 2).

Anchor-binding mAbs were broadly reactive and broadly neutralizing to pre-pandemic and post-pandemic H1N1 viruses and a swine-origin H1N2 virus (Fig. 1e, Extended Data Fig. 1g, h). Notably, anchor-binding mAbs had similar neutralizing potency to pH1N1 relative to mAbs to the CS epitope (Fig. 1e, Extended Data Fig. 1i, j). Many stalk-targeting antibodies mediate protection via Fc-mediated functions, including antibody-dependent cellular cytotoxicity[15,16]. Anchor epitope-binding mAbs largely did not possess antibody-dependent cellular cytotoxicity activity (15 out of 18; Extended Data Fig. 1k–l), potentially due to the upward angle of approach of anchor-binding antibodies, which may position the Fc distantly from effector cells. Despite pan-H1 binding, anchor-targeting mAbs rarely cross-reacted with other HA subtypes tested, including H3, a group 2 subtype, other group 1 subtypes, including H2 and H5, and influenza B viruses (Fig. 1f, Extended Data Fig. 2a). Despite this, the 222-1C06 mAb cross-reacted with H2 and H5 HA (Fig. 1g, Extended Data Fig. 2b) and several anchor-binding antibodies could neutralize an H2N2 virus (Extended Data Fig. 2c), suggesting that antibodies targeting the anchor epitope can cross-neutralize other group 1 influenza A viruses. We recently demonstrated that bnAbs to the HA stalk are often polyreactive[17], which may limit the activation of B cells expressing bnAbs. Relative to mAbs that target the CS epitope, we identified that anchor-binding mAbs were proportionally less likely to be polyreactive and those that were polyreactive had weaker relative affinity for lipopolysaccharide (Extended Data Fig. 2d, e). These data suggest that, although polyreactivity is selected for within the anti-anchor epitope B cell pool, it is not to the same level as B cells to the CS epitope.

H1N1 viruses have acquired several mutations within the HA stalk domain, probably due to antibody selective pressures or to increase stability. To understand whether these mutations affect antibody binding to the anchor epitope, we screened mAbs against naturally occurring and experimentally identified viral escape mutants of mAbs binding to the CS epitope (Extended Data Fig. 2f, g, Extended Data Table 3). Anchor epitope-binding mAbs were mostly unaffected by these mutants, whereas most of the CS-binding mAbs showed reduced binding to at least one mutant (Extended Data Fig. 2g). Notably, most mAbs had reduced binding to A373V of HA2, which has recently been shown to preferentially grow in the presence of mAbs to the CS epitope[2]. While A373 is distant from the anchor epitope, the A373V mutation has been shown to affect the conformation of the HA stalk[2], explaining the broad reduction of HA binding by antibodies targeting either stalk epitope. Anchor-binding mAbs only demonstrated a 10–30% reduction in binding (Extended Data Fig. 2g), indicating that they are still likely to neutralize viruses carrying the A373V mutation.

**Fig. 1 | The anchor epitope is a common target of stalk-binding antibodies.** **a**, Negative-stain EM of representative 2D class averages and 3D reconstructions of Fabs binding to A/California/7/2009 (E376K) HA. 047-09 4F04 was imaged at ×52,000 normal magnification and 222-1C06 and 241 IgA 2F04 were imaged at ×62,000 normal magnification. **b**, Juxtaposed 3D reconstructions of Fabs binding to A/California/7/2009 (E376K) HA. **c**, Binding footprints of anchor-binding Fabs relative to mAbs targeting the CS epitope (CR9114 and FI6v3). **d**, Competition of stalk-binding mAbs with CR9114 or 047-09 4F04 (bold mAbs). **e**, Neutralization potency of anchor-binding (*n* = 50) and CS-binding (*n* = 37) mAbs to A/California/7/2009. Data are represented as mean ± s.d. and were analysed by a two-tailed unpaired non-parametric Mann–Whitney test. IC$_{50}$, half-maximal inhibitory concentration. **f**, Proportion of anchor-targeting (*n* = 50) and CS-targeting (*n* = 37) mAbs binding to other group 1 influenza virus A subtypes. Data were analysed by Fisher's exact tests. **g**, Representative 2D class averages (×62,000 normal magnification), 3D reconstructions and footprints of 222-1C06 binding to H2 and the relative footprint on H1. See also Extended Data Figs. 1–3.

central stalk (CS) epitope (Extended Data Fig. 1a). Notably, stalk-binding mAbs were disproportionally isolated from the infected, 2009 pandemic H1N1 (pH1N1) monovalent inactivated influenza vaccine, and the cHA vaccine cohorts (Extended Data Fig. 1b), as these exposure routes have previously been shown to induce antibody responses to the HA stalk[5,8,9]. To investigate which epitopes the remaining 79% of stalk-binding mAbs were targeting, we performed negative-stain electron microscopy with three non-CR9114 (ref. [10]) competing stalk domain-binding mAbs: 047-09 4F04, 241 IgA 2F04 and 222-1C06. All

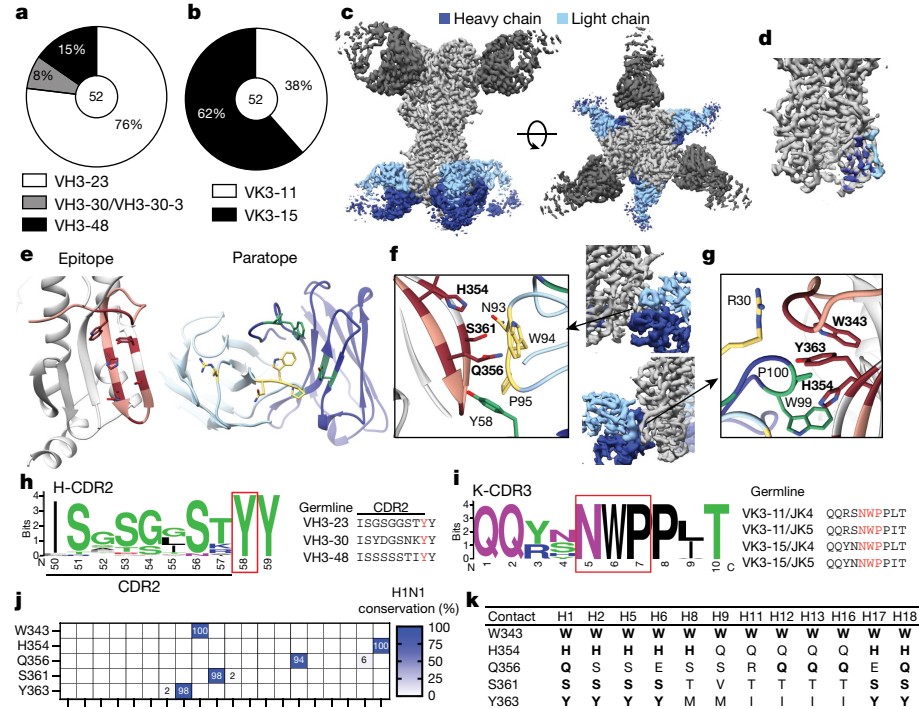

**Fig. 2 | Anchor-targeting mAbs bind to the HA fusion peptide via public binding motifs. a**, **b**, VH (**a**) and VK (**b**) gene usage by anchor-binding mAbs. The number in the centre of the pie graphs indicates the number of mAbs analysed. **c**, Cryo-EM structure of anchor-targeting 222-1C06 (blue) and lateral patch-targeting 045-09 2B05 (dark grey; see Methods) binding to A/California/7/2009 (E376K) HA (light grey). **d**, Heavy chain and light chain footprint of 222-1C06 binding to HA based on the cryo-EM structure. **e**, HA epitope contact residues (maroon) and heavy chain (green) and light chain (yellow) antibody contact residues of the 222-1C06 paratope. Peach-highlighted amino acids represent the fusion peptide of HA2. **f**, K-CDR3 NWP and H-CDR2 Y58 motifs of 222-1C06. Bold residues are HA residues. **g**, Major contacts of 222-1C06 K-CDR1 and H-CDR3 (normal typeface) binding to HA (bold residues). **h**, **i**, Weblogo plot and germline sequence of Y58 following the H-CDR2 motif (**h**) and the K-CDR3 NWP motif (**i**). **j**, **k**, Amino acid conservation of s contact residues across human, swine and avian H1 viruses (**j**) and group 1 influenza A viruses (**k**). Bold residues are contacts conserved with A/California/7/2009 H1N1 (**k**). The strain information used for conservation models in **j** and **k** are in Supplementary Tables 4, 5, respectively. See also Extended Data Fig. 4.

To test whether mAbs targeting the anchor epitope were protective in vivo, we administered a cocktail of five mAbs targeting the anchor epitope or the CS epitope prophylactically and therapeutically (48 h after infection) to mimic a polyclonal response against these epitopes and infected mice with a lethal dose of a mouse-adapted pH1N1 virus (A/Netherlands/602/2009; Supplementary Table 1). Mice that received a prophylactic or therapeutic anti-anchor cocktail at 5 mg/kg experienced 100% protection from weight loss and lethal infection (Extended Data Fig. 3a, b). No differences in lung viral titres were detected in mice that received the anti-anchor cocktail prophylactically relative to mice administered the negative control mAb (Extended Data Fig. 3c). Anti-stalk antibodies do not provide sterilizing immunity but are known to neutralize subsequent rounds and limit disease progression[18]. As a result, lung viral titres do not necessarily reflect protection from morbidity and mortality. Finally, the anti-anchor cocktail given prophylactically provided 70% protection against lethal A/Fort Monmouth/1/1947 infection (Extended Data Fig. 3d), a virus that circulated before the birth of any of the donors in our study (Extended Data Table 1). Since anchor-binding mAbs do not engage in antibody-dependent cellular cytotoxicity for the most part, antibodies that target the anchor epitope probably provide protection in vivo through direct neutralization of the virus. Together, these data indicate that antibodies to the anchor epitope are pan-H1 neutralizing and protective in vivo.

## Structure of an anchor-binding antibody

All anchor epitope-binding mAbs utilized one of four VH3 genes: VH3-23, VH3-30 or VH3-30-3, and VH3-48, in contrast to mAbs targeting the CS epitope, which commonly use VH1-69 (Fig. 2a, Extended Data Fig. 4a, b). Anchor epitope-binding mAbs also utilized a highly restricted light chain repertoire, with all mAbs utilizing a combination of VK3-11 or VK3-15 combined with JK4 or JK5 (Fig. 2b, Extended Data Fig. 4c, d). Furthermore, all but one light chain of the anchor-targeting mAbs were clonally related (Extended Data Table 2), indicating that the light chains were very similar across mAbs and study participants. We identified four distinct clonal expansions, with one public clone found across two donors (Extended Data Fig. 4e, f, Extended Data Table 2). Anchor epitope-targeting and CS-targeting mAbs exhibited similar levels of somatic hypermutations (Extended Data Fig. 4g). The κ-chain complementarity-determining region 3 (K-CDR3) length of anchor epitope-binding mAbs was highly restricted, with all K-CDR3s being ten amino acids long (Extended Data Fig. 4h).

To investigate the binding motif of anchor-targeting mAbs, we generated a high-resolution (3.38 Å) cryo-electron microscopy (cryo-EM) structure of 222-1C06 bound to A/California/7/2009 HA (Fig. 2c, Extended Data Fig. 4i–k). The broad paratope of 222-1C06 made extensive contacts across the HA fusion peptide[19] (Fig. 2d, e), which mediates viral membrane fusion with the host membrane. Binding of the fusion peptide was largely mediated by an NWP motif within the K-CDR3, a Y58 directly following the H-CDR2, and a W99 in the H-CDR3, with these HA-binding motifs acting independently and in combination via an aromatic pocket (Fig. 2f, g, Supplementary Table 2). Moreover, both the K-CDR3 NWP and the H-CDR2 Y58 were found in all the anchor-binding mAbs and were germline encoded (Fig. 2h, i), which could have led to the selection of B cells utilizing these variable genes. Notably, FISW84 utilizes VH3-23 and VK3-15, and comparison of the paratopes showed that the NWP and Y58 motifs of FISW84 similarly form an aromatic pocket but orient towards the fusion peptide slightly differently than

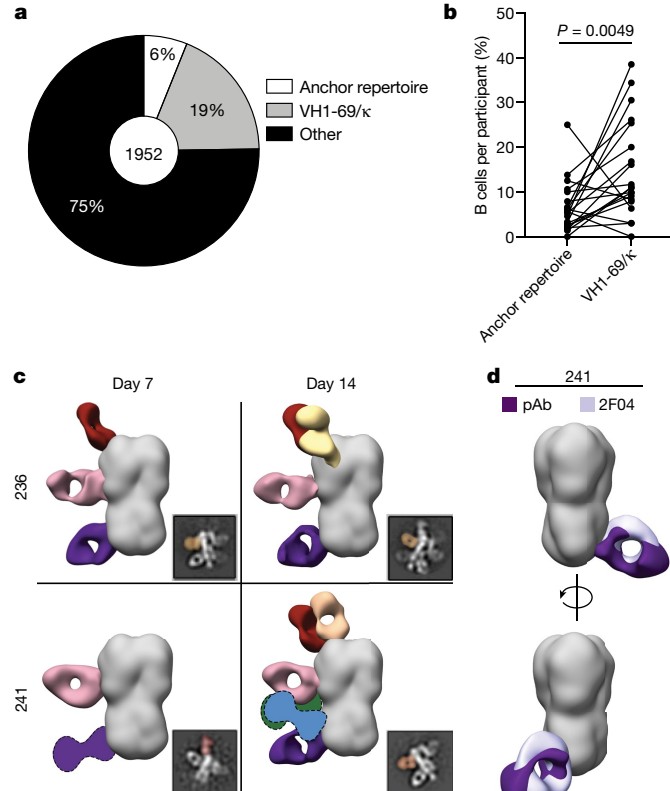

**Fig. 3 | MBCs and serum antibodies commonly target the anchor epitope. a**, Proportion of all cH5/1+ B cells with repertoire features of anchor-binding mAbs, VH1-69/κ (CS epitope) or with other repertoire features. The number in the centre of the pie graph is the number of B cells analysed. **b**, Proportion of B cells with anchor-binding mAb features or that use VH1-69/κ-chain by participant (*n* = 20 donors). Lines connect the same participant. Data were analysed using a two-tailed paired non-parametric Wilcoxon matched-pairs signed rank test. **c**, Electron microscopy polyclonal epitope mapping (EMPEM) summary of polyclonal antibodies (pAbs) binding to A/Michigan/45/2015 HA in the serum of participants 236 and 241 collected at day 7 and day 14 following 2014 quadrivalent inactivated influenza vaccine. Fabs shown as graphics with dotted lines represent predicted placements due to limited particle representation. 2D class averages were imaged at ×62,000 normal magnification **d**, Overlap of 241 IgA 2F04 Fab and pAb binding the anchor epitope from participant 241. See also Extended Data Fig. 5.

222-1C06 (Extended Data Fig. 4l). Molecular dynamics simulations also showed that VH3-23-utilizing and VH3-30-utilizing mAbs from our study and FISW84 targeted the HA fusion peptide similarly to the cryo-EM structure of 222-1C06 via the aromatic pocket created by the K-CDR3 NWP and H-CDR2 Y58 motifs, albeit at different orientations (Extended Data Fig. 4m). Crucially, molecular dynamic and cryo-EM analyses revealed numerous intra-Fab interactions of hydrophobic and aromatic amino acids, including p-stacking of K-CDR3-P95 with K-CDR3-W94 and H-CDR2-Y58 that rigidified the paratope (Extended Data Fig. 4n, o, Supplementary Table 3).

Broad analysis of human, swine and avian H1 viruses revealed that the side-chain contacts of 222-1C06 were highly conserved (94–100% conserved; Fig. 2j). In addition, the side-chain contacts were 100% conserved across 100 years of H1N1 virus evolution in humans (Extended Data Fig. 4p). Deep mutational scanning of potential H1 viruses at these contact residues indicated substantial permissibility (Extended Data Fig. 4q), although these mutations appear to not have been selected for in nature (Fig. 2j). The five side-chain contacts of this broad epitope were also highly conserved across group 1 viruses, with the W343 contact being

100% conserved across all group 1 viruses (Fig. 2k). Together, these data reveal that B cells targeting the anchor epitope utilized a highly restricted V(D)J gene repertoire, and the specific features within this repertoire made critical and extensive contacts with the conserved anchor epitope.

## The anchor epitope is a common target

Owing to the restricted repertoire features of anchor-targeting mAbs, we next determined the relative proportion of B cells with these features by interrogating single-cell repertoire sequencing of HA-specific B cells isolated from 20 human participants following cHA vaccination[4,5]. The cHA vaccine platform is intended to specifically induce antibodies to the stalk domain by retaining the stalk domain of H1 and replacing the head domain of HA with that of an avian subtype, either H8 (prime) or H5 (boost) for this trial[4,5]. To investigate the proportion of B cells with anchor epitope-binding repertoire features, we selected B cells that used VH3-23/VH3-30/VH3-30-3/VH3-48, VK3-11/VK3-15, JK4/JK5, a K-CDR3 ten amino acids in length, and possessed an NWP motif within the K-CDR3. We also segregated B cells expressing VH1-69 and a κ-chain, which are commonly used by B cells that target the CS epitope. We identified that B cells with features of antibodies binding the anchor epitope were abundant within the human B cell repertoire, with 6% of all B cells identified fitting within this defined repertoire (Fig. 3a). Moreover, all but one participant had at least one B cell with anchor-binding repertoire features (Fig. 3b). In addition, 32 out of 33 mAbs generated from the selected anchor-targeting B cell list competed for HA binding with 047-09 4F04 (Extended Data Fig. 5a), indicating that this population was greatly enriched for anchor-targeting B cells. Moreover, the anchor-binding B cells were highly mutated and were largely class-switched to IgG1 (Extended Data Fig. 5b, c), suggesting that these B cells were MBCs that had undergone affinity maturation and class-switch recombination. Together, these data indicate that the anchor epitope is a common target of the human MBC repertoire following cHA vaccination.

To confirm that anchor epitope-targeting mAbs were representative of the serum antibody response, we performed electron microscopy polyclonal epitope mapping (EMPEM)[20] with serum antibodies from participants 236 and 241 from the 2014 quadrivalent inactivated influenza vaccine cohort (Extended Data Table 1). Both participants had detectable antibodies targeting the anchor epitope at days 7 and 14 post-vaccination (Fig. 3c, Extended Data Fig. 5d, e). Comparison of anchor epitope-binding polyclonal antibodies identified in participant 241 revealed that the 241 IgA 2F04 mAb strongly overlapped with the 241 polyclonal antibody (Fig. 3d), suggesting that the polyclonal antibody derived from this clonal expansion. Together, these data indicate that humoral immunity against the anchor epitope is common within the MBC pool and polyclonal serum antibody response after vaccination.

## cHA induces anchor-binding antibodies

To investigate whether participants enrolled in a phase I clinical trial of the cHA vaccine (Fig. 4a) mounted an antibody response to the anchor and the CS epitope, we adapted the competition ELISA to detect serum antibody responses that could compete for binding with 047-09 4F04 and CR9114, respectively. Three different vaccine formulations were used in this trial, with participants being primed with either a cH8/1 inactivated influenza vaccine (IIV) with an oil-in-water adjuvant (AS03) or a cH8/1 live-attenuated influenza vaccine followed by a boost with a cH5/1 IIV with or without AS03 (Fig. 4a). Only participants who received the IIV + AS03 on either the prime or boost were capable of seroconverting against both the anchor and the CS epitopes (Fig. 4b, c, Extended Data Fig. 6a, b). Participants who received the cH8/1 IIV + AS03 prime did not further increase serum antibodies to either stalk epitope after the cH5/1 + AS03 boost (Fig. 4c), suggesting that these B cells were refractory to continued activation. Serum titres against the anchor and CS epitopes (Extended Data Fig. 6a, b) closely matched that of serum

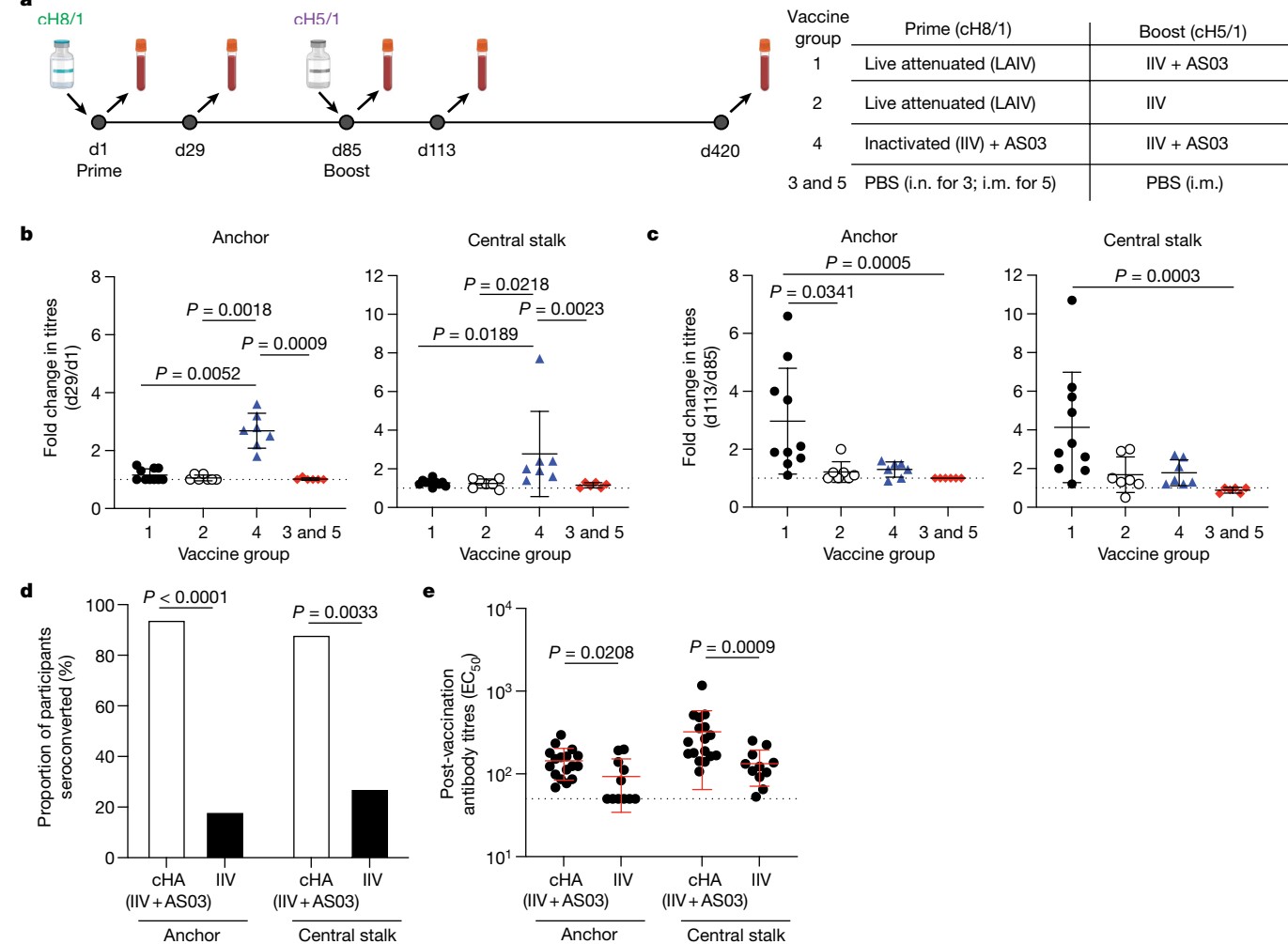

**Fig. 4 | cHA vaccination in humans robustly recalls MBCs targeting the anchor epitope. a**, cHA vaccine trial design including vaccine groups (right; group 1 *n* = 10 participants; group 2 *n* = 7 participants; group 4 *n* = 7 participants; group 3 and 5 *n* = 6). i.m., intramuscular; i.n., intranasal; LAIV, live-attenuated influenza vaccine; PBS, phosphate buffered saline. Bottle images created with BioRender.com. **b**, **c**, Fold change by participant of serum antibodies competing for binding to the anchor and CS epitopes after the prime (d29/d1; **b**) and the boost (d113/85; **c**). **d**, **e**, Proportion of participants who seroconverted (**d**) and half-maximal effective concentration titres (EC$_{50}$; **e**) to the anchor and CS epitopes. Individuals in the cHA (IIV + AS03) cohort were those who received the cHA vaccine with adjuvant (cH8/1 IIV + AS03 prime and cH5/1 IIV + AS03 boost; *n* = 17 donors) and the IIV cohort were those who received licensed IIV vaccines (2009 monovalent inactivated influenza vaccine, 2010 trivalent inactivated influenza vaccine and 2014 quadrivalent inactivated influenza vaccine; *n* = 11 donors). Data in **b**, **c** and **e** are mean ± s.d. The dotted line represents the limit of detection. Data in **b** and **c** were analysed by two-tailed non-parametric Kruskal–Wallis tests. Data in **d** were analysed by Fisher's exact test. Data in **e** were analysed by two-tailed unpaired non-parametric Mann–Whitney test. See also Extended Data Fig. 6.

neutralizing titres against a cH6/1N8 virus and an avian-swine H1N1 virus[5], suggesting that the anchor-targeting and CS-targeting serum antibodies were responsible for neutralization.

All but one participant in the IIV + AS03 groups seroconverted against the anchor epitope (Fig. 4d) and had higher titres against the anchor epitope than participants who received the 2009 monovalent inactivated influenza vaccine or seasonal influenza virus vaccines (Fig. 4e). However, the precise role of the cHA immunogen versus the AS03 adjuvant in inducing anti-stalk antibody responses could not be resolved in our study. Notably, those individuals who received the IIV alone had weak plasmablast responses relative to those individuals who received the IIV + AS03 (ref. [4]), suggesting that the oil-in-water adjuvant, not the cHA immunogen, was essential for robust activation of B cells and anti-anchor antibody responses. Moreover, considerable pre-existing antibodies may hinder recall of B cells to the stalk. Individuals first exposed to the 2009 pH1N1, a virus for which

individuals had low pre-existing humoral immunity, had proportionally more isolated mAbs to the stalk and were more likely to have an anti-anchor mAb than individuals who had repeatedly been exposed to the pH1N1 virus in subsequent influenza seasons (Extended Data Fig. 6c, d), therefore suggesting that pre-existing antibodies may limit antibody responses to the HA stalk. Despite robust recall of antibodies to the anchor and CS epitopes by the adjuvanted vaccines, titres decreased 1 year after vaccination (day 420; Extended Data Fig. 6e, f). Together, these data indicate that the adjuvanted inactivated cHA vaccine can robustly induce antibodies to multiple stalk epitopes, including the anchor.

Headless HA antigens, including mini-HA[21], are attractive universal influenza virus vaccine antigens, as these antigens lack the immunodominant epitopes of the HA head[21,22]. However, only 1 out of 50 anchor-binding mAbs bound the mini-HA antigen[21], whereas all anchor epitope-binding mAbs bound cH6/1 (Extended Data Fig. 6g).

Compared to full-length HA, the membrane-proximal region of the mini-HA splays by approximately 14.5 Å (ref. [21]), which may disrupt the antigenicity of the anchor epitope. To understand whether anchor epitope-targeting antibodies could bind to the mini-HA in a more native setting, we generated a membrane-bound mini-HA and observed that mAbs targeting the anchor and CS epitopes readily bind to both the full-length membrane-bound A/California/7/2009 HA and the membrane-bound mini-HA (Extended Data Fig. 6h), indicating that the mini-HA is antigenic when natively presented. Furthermore, we demonstrated that anchor epitope-targeting antibodies bound HA with a fibritin but not a GCN4 trimerization domain (Extended Data Fig. 6i), highlighting selection of the trimerization domain as an important factor for vaccine design. Together, these data demonstrate that native-like HA antigens, such as the cHA vaccine, can recall MBCs that target the anchor epitope.

## Discussion

In this study, we identified a public class of bnAbs that target an epitope at the anchor of the HA stalk domain near the membrane. The anchor-targeting mAbs were public clonotypes across participants, with all antibodies possessing two conserved, germline-encoded binding motifs: a Y58 directly following the H-CDR2 and an NWP motif within the K-CDR3. The neutralizing activity of anchor epitope-targeting mAbs to pre-pH1N1 and post-pH1N1 viruses and a swine-origin H1-expressing virus indicates that the anchor epitope is an important target for pan-subtype neutralizing antibodies. As two of the last four influenza virus pandemics were caused by H1N1 viruses and a recent report has shown that antigenically novel H1-expressing viruses commonly spill over from swine into humans[23], it is critically important to generate pan-H1 vaccines to prevent the next influenza pandemic. Moreover, the ability of anti-anchor antibodies to neutralize an H2 virus and the general conservation of contact residues suggests that anchor-targeting antibodies have the potential to acquire cross-neutralization against group 1 viruses.

Our study highlights an additional broadly protective epitope of the HA stalk and provides guidance on how vaccines can be designed to drive broadly protective antibodies to multiple distinct epitopes, which can work cooperatively to provide optimal protection while avoiding the generation of antibody escape mutants. Notably, studies in the HIV-1 field have shown that bnAb monotherapy can lead to the development of antibody-resistant viral variants[24–26], whereas combination bnAb therapy does not[27]. In addition, immune focusing towards a single epitope can lead to the generation and selection of viral escape mutants at these highly conserved epitopes[2,3,28,29]. Therefore, it is critical that future universal influenza virus vaccines elicit antibodies to multiple conserved epitopes to prevent the generation of bnAb escape viruses.

The angle of approach of anchor-binding mAbs and the proximity of the epitope to the viral membrane suggest that this epitope is typically obstructed, limiting antibody recognition and B cell activation. However, membrane-bound HA typically flexes between 0° and 25° and up to 52° on its threefold axis[11], suggesting antibodies and B cells can easily access the anchor epitope during this flexing process. Moreover, H1 viruses possess a highly conserved glycosylation site on the HA stalk that lies above the anchor epitope[11,30], which may obstruct antibodies from targeting this epitope from above. Similarly, a neutralizing mAb to the Middle East respiratory syndrome coronavirus targets an epitope on S2 from an upward angle to avoid glycans[31]. Therefore, the upward angle of approach may be a common feature of antibodies that recognize epitopes below glycans.

Our study shows that humans have pre-existing immunity against the anchor epitope and influenza virus vaccination can recall MBCs to secrete antibodies to this epitope. However, vaccine HA antigens must have a native-like conformation near the transmembrane domain, as trimer splaying potentially due to the GCN4 trimerization domain ablates antibody binding to the anchor epitope. Moreover, our study

reveals that the cHA vaccine strategy recalled MBCs to the anchor and CS epitopes, as these MBCs do not need to compete against MBCs that target the immunodominant variable HA head epitopes[3,9,17,32]. Similarly, the mini-HA/headless HA vaccine strategy has the potential to also recall MBCs to multiple epitopes of the HA stalk domain, if displayed natively[22,33]. The addition of an oil-in-water AS03 adjuvant to the cHA was critical for recalling MBCs to the anchor epitope. Oil-in-water adjuvants largely function to emulsify antigen, which may prevent sequestration of antigen by circulating antibodies, increase delivery of free antigen to lymph nodes, and help to stimulate innate immune receptors[34,35]. Notably, an oil-in-water adjuvanted inactivated H5N1 vaccine induced higher neutralizing antibody titres, antibodies to more HA epitopes, and higher avidity antibodies[36]. Therefore, the inclusion of oil-in-water adjuvants may have an important role in generating bnAbs and may improve vaccine effectiveness of seasonal influenza vaccines. Together, our study shows that influenza vaccination strategies, such as the cHA vaccine with the AS03 adjuvant, have the capability to robustly induce antibodies to the previously underappreciated anchor epitope and can provide broad protection against H1 viruses.

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

# Methods

## Study approvals, cohorts and human materials

Human peripheral blood mononuclear cells (PBMCs) and serum were obtained from multiple donors from multiple cohorts, which is outlined in Extended Data Table 1. Informed consent was obtained from all participants. All studies were performed with the approval of the University of Chicago Institutional Review Board (IRB; ID #09-043-A). The 2009 pH1N1 infection and 2009 monovalent inactivated influenza vaccine (MIV) cohorts were also approved by the IRB at Emory University. The chimeric HA vaccine study cohort is identified as clinical trial NCT03300050 and further details on trial design are outlined elsewhere[4,5]. The study was approved by IRBs at local clinical sites, including Icahn School of Medicine at Mount Sinai, Duke University, and Cincinnati Children's Hospital Medical Center. All experiments performed with mice were done in accordance with the University of Chicago and Icahn School of Medicine at Mount Sinai Institutional Animal Care and Use Committees.

## Cell lines

Human embryonic kidney HEK293T (female, #CRL-11268), Madin Darby canine kidney (MDCK; female, #CCL-34, NBL-2) and human A549 (#CCL-185) cells were purchased and authenticated by the American Type Culture Collection (ATCC). MDCK-SIAT1 cells were generated previously[37]. All cells were maintained in a humidified atmosphere of 5% $CO_2$ at 37 °C. HEK293T cells were maintained in advanced-DMEM supplemented with 2% ultra-low IgG fetal bovine serum (FBS; Invitrogen), 1% L-glutamine (Invitrogen) and 1% antibiotic-antimycotic (Invitrogen). MDCK, MDCK-SIAT1 and A549 cells were maintained in DMEM supplemented with 10% FBS (Invitrogen), 1% L-glutamine (Invitrogen) and 1% penicillin–streptomycin (Invitrogen). Jurkat cells expressing FcgRIIIa and FcgRI with a NFAT-driven luciferase reporter gene (#G7010) were acquired and validated by Promega and were directly used for antibody-dependent cellular cytotoxicity (ADCC) assays. Cell lines were not authenticated after receiving from suppliers and were not tested for mycoplasma.

## Monoclonal antibody production

Monoclonal antibodies were generated as previously described[38–40]. Peripheral blood was obtained from each donor approximately 7 days after vaccination or infection or obtained 28[+] days post-vaccination. Lymphocytes were isolated and enriched for B cells using RosetteSep. Total plasmablasts (CD3−CD19+CD27hiCD38hi; all cohorts except 2014 quadrivalent inactivated influenza vaccine (QIV)), IgG+ plasmablasts (CD3−CD19+IgM−CD27hiCD38hiIgG+IgA−; 2014 QIV), IgA+ plasmablasts (CD3−CD19+IgM−CD27hiCD38hiIgG−IgA+; 2014 QIV cohort), or HA+ bait-sorted MBCs (CD3−CD19+CD27+CD38lo/+HA+; for 030-09M 1B06) were single-cell sorted into 96-well plates. Genes encoding immunoglobulin heavy and light chains were amplified by reverse transcriptase PCR (RT–PCR), sequenced, cloned into human IgG1, human κ-chain, or human λ expression vectors, and co-transfected into HEK293T cells. Secreted mAbs were purified from the supernatant using protein A agarose beads. For mAbs generated from the 2014 QIV cohort, mAb names include the original isotype of the sorted plasmablast, and all mAbs were expressed as human IgG1. cH5/1-binding B cells (CD19+CD27+cH5/1+) were sorted from donors 28 days after cH5/1 vaccination (NCT03300050). Cells were sorted with A/California/04/2009 HA probe (for 030-09M 1B06) or cH5/1 probe, each with a Y98F mutation to ablate non-specific binding to sialic acids on B cells. mAb heavy chain and light chain sequences were synthesized from single-cell RNA sequencing data of cH5/1-baited B cells (IDT), and cloned into the human IgG1, human κ-chain or human λ expression vectors. B cell clones were determined by aligning all the V(D)J sequences sharing identical progenitor sequences, as predicted by IgBLAST using our in-house software, VGenes. Consensus sequence analysis was performed using WebLogo[41] and sequence alignments were determined using Clustal Omega.

## Viruses and recombinant proteins

Influenza viruses used in all assays were grown in-house in specific pathogen free (SPF) eggs, harvested, purified and titred. Recombinant HA, cHA and mini-HA were obtained from BEI Resources or generated in-house. Recombinant HA mutant proteins used in Extended Data Fig. 2 were generated with identified mutations from the deep mutational scanning experiments (see below) or with known mutations that have arisen naturally or were identified in other studies[2,10,42–52] (Extended Data Table 3). All mutations were made on HA from A/California/7/2009 and were expressed in HEK293T cells and purified using Ni-NTA agarose beads (Qiagen).

## Antigen-specific ELISA

High protein-binding microtitre plates (Costar) were coated with 8 haemagglutination units (HAU) of virus in carbonate buffer or with recombinant HA, including HA mutants described below, at 2 μg ml[−1] in phosphate-buffered saline (PBS) overnight at 4 °C. Plates were washed the next morning with PBS 0.05% Tween and blocked with PBS containing 20% FBS for 1 h at 37 °C. Antibodies were then serially diluted 1:3 starting at 10 μg ml[−1] and incubated for 1.5 h at 37 °C. Horseradish peroxidase (HRP)-conjugated goat anti-human IgG antibody diluted 1:1,000 (Jackson Immuno Research) was used to detect binding of mAbs, and plates were subsequently developed with Super Aquablue ELISA substrate (eBiosciences). Absorbance was measured at 405 nm on a microplate spectrophotometer (Bio-Rad). To standardize the assays, control antibodies with known binding characteristics were included on each plate, and the plates were developed when the absorbance of the control reached 3.0 optical density (OD) units. All ELISAs were performed in duplicate twice. To define antibodies as targeting the H1 stalk, mAbs were tested for binding to cH5/1, which utilizes the head domain from H5-expressing viruses and the stalk domain from the pH1N1 virus[53], and for haemagglutination inhibition (HAI) activity against pH1N1 (A/California/7/2009). mAbs that bound the cHA and lacked HAI activity were classified as those binding the HA stalk domain. To classify antigen specificity, mAbs that did not definitively bind to the HA head or stalk are listed as binding unknown HA+ epitopes. Affinity measurements, as represented as dissociation constant ($K_d$) at a molar concentration (M), were calculated using Prism 9 (GraphPad) by performing a non-linear regression. All experiments were performed in duplicate and technically replicated twice.

## Competition ELISAs

Plates were coated with 50 μl of A/California/7/2009 HA at a concentration of 1 μg ml[−1] and incubated overnight at 4 °C. To biotinylate the antibodies with known epitope specificities, CR9114 (CS epitope) and 047-09-4F04 (anchor epitope) were incubated at 4 °C with EZ-Link Sulfo-NHS-Biotin (Thermo Scientific) for 24 h or 48 h before use, respectively. After blocking the plates with PBS containing 20% FBS for 1 h at 37 °C, serum samples were incubated (starting dilution of 1:50 for human serum or 20 μg ml[−1] for mAbs) in the coated wells for 2 h at room temperature. Either biotinylated CR9114 or 047-09 4F04 was then added at a concentration equal to twice its $K_d$ and incubated in the wells with the serum or mAbs for 2 h at room temperature. The biotinylated antibodies were desalted before addition to remove free biotin using Zeba spin desalting columns, 7 k MWCO (Thermo Scientific). After washing the plates, wells were incubated with HRP-conjugated streptavidin (Southern Biotech) at 37 °C for 1 h for detection of the biotinylated antibody. Super Aquablue ELISA substrate (eBiosciences) was then added, and absorbance was measured at 405 nm on a microplate spectrophotometer (Bio-Rad). To standardize the assays, biotinylated CR9114 or 047-09 4F04 was incubated in designated wells on each plate without any competing serum or mAb, and data were recorded when

the absorbance of these wells reached an OD of 1–1.5 units. After subtracting background, percent competition by serum samples was then determined by dividing the observed OD of a sample by the OD reached by the positive control, subtracting this value from 1, and multiplying by 100. For the serum data, ODs were log transformed and analysed by non-linear regression to determine $EC_{50}$ values using Prism software (GraphPad). For Fig. 4 and Extended Data Fig. 5, only donors with serum for all timepoints were included. All experiments were performed in duplicate and technically replicated twice.

## Polyreactive ELISAs

High-protein binding microtitre plates (Costar) were coated with 10 µg ml$^{-1}$ calf thymus double-stranded DNA (dsDNA; Thermo Fisher Scientific), 2 µg ml$^{-1}$ *Salmonella enterica* serovar Typhimurium flagellin (Invitrogen), 5 µg ml$^{-1}$ human insulin (Sigma-Aldrich), 10 µg ml$^{-1}$ KLH (Invitrogen) and 10 µg ml$^{-1}$ *Escherichia coli* LPS (Sigma-Aldrich) in PBS. Plates were coated with 10 µg ml$^{-1}$ cardiolipin in 100% ethanol and allowed to dry overnight. Plates were washed with water and blocked with PBS, 0.05% Tween and 1 mM EDTA. mAbs were diluted 1 µg ml$^{-1}$ in PBS and serially diluted fourfold and added to plates for 1.5 h. Plates were washed and goat anti-human IgG-HRP (Jackson Immunoresearch) was diluted 1:2,000 in PBS, 0.05% Tween and 1 mM EDTA. Plates were washed with water and were blocked with PBS, 0.05% Tween and 1 mM EDTA for 5 min. Plates were washed again with water and were developed with Super Aquablue ELISA substrate (eBioscience) until the positive control mAb, 3H9 (ref. [54]), reached an $A_{450}$ of 3. All experiments were performed in duplicate and technically replicated twice.

## Deep mutational scanning for stalk domain mutants

The mutant libraries used herein were previously described[55]. The libraries consist of all single amino acid mutations to A/WSN/1933 (H1N1). The experiments were performed by using biological triplicate libraries. The mutational antigenic profiling of 045-09 2B06, a CS epitope-binding mAb, was performed as previously outlined[56]. In brief, $10^6$ tissue culture infectious dose 50 ($TCID_{50}$) of two of the virus library biological replicates was diluted in 1 ml in IGM (Opti-MEM supplemented with 0.01% FBS, 0.3% BSA and 100 mg ml$^{-1}$ calcium chloride) and incubated with an equal volume of 045-09 2B06 antibody at a final concentration of 50 or 25 µg ml$^{-1}$ for 1.5 h at 37 °C. MDCK-SIAT1 cells were infected with the virus antibody mixtures. Two hours post-infection, the media were removed, the cells washed with 1 ml PBS, and 2 ml of fresh IGM was added. Fifteen hours post-infection, viral RNA was extracted, reverse-transcribed using primers WSNHA-For (5′-AGCAAAAGCAGGGGAAAATAAAAACAAC-3′) and WSNHA-Rev (5′-AGTAGAAACAAGGGTGTTTTTCCTTATATTTCTG-3′), and PCR amplified according to the barcoded-subamplicon library preparation as previously described[55]. The overall fraction of virions that survive antibody neutralization was estimated using quantitative RT–PCR (qRT–PCR) targeting the viral nucleoprotein (NP) and cellular GAPDH as previously described[56]. Using tenfold serial dilutions of the virus libraries, we infected cells with no antibody selection to serve as a standard curve of infectivity. qPCR Ct values from the standard curve samples compared to the virus–antibody mix samples were determined for NP and GAPDH. We then generated a linear regression to fit the difference between the NP and GAPDH Ct values for the standard curve samples, and then used this curve to interpolate the fraction surviving for the antibody–virus selection samples. Across the three library replicates, the fraction of virus surviving antibody selection was 0.17, 0.1 and 0.14.

Illumina(R) deep sequencing data were analysed using dms_tools2 version 2.4.12 software package[57], which can be found at https://github.com/jbloomlab/dms_tools2. The computer code used is at https://github.com/jbloomlab/2B06_DMS, and the Jupyter notebook that performed most of the analysis is at https://github.com/jbloomlab/2B06_DMS/blob/master/analysis_notebook.ipynb. The sequencing counts were processed to estimate the differential selection for each mutation, which is the log enrichment of that mutation in the antibody-selected condition versus the control[56]. The numerical measurements of the differential selection that 2B06 imposes on each mutation can be found at: https://github.com/jbloomlab/2B06_DMS/blob/master/results/diffsel/tidy_diffsel.csv.

## Deep mutational scanning for H1 variants

Amino acid preferences for the HA of A/WSN/1933 (H1N1) were previously determined[55]. In brief, deep mutational scanning was performed by passaging virus libraries at a low multiplicity of infection in MDCK-SIAT1 cells. Following deep sequencing of the resulting virus, amino acid preferences were determined using the Python package dms_tools (http://jbloomlab.github.io/dms_tools/), version 1.1.12. This program aligns subamplicon reads to the reference HA sequence, counts the number of mutations at each amino acid site, and determines amino acid preferences based on the mutation counts pre-selection and post-selection.

## Microneutralization assays

Microneutralization assays for mAb characterization were carried out as previously described[58,59]. MDCK cells were maintained in DMEM supplemented with 10% FBS, 1% penicillin–streptomycin, and 1% L-glutamine at 37 °C with 5% $CO_2$. The day before the experiment, 25,000 MDCK cells were added to each well of a 96-well plate. Serial twofold dilutions of mAb were mixed with an equal volume of 100 $TCID_{50}$ of virus for 1 h and added to MDCK cells for 1 h at 37 °C. The mixture was removed, and cells were cultured for 20 h at 37 °C with 1X MEM supplemented with 1 µg ml$^{-1}$ tosyl phenylalanyl chloromethyl ketone (TPCK)-treated trypsin and appropriate mAb concentration. Cells were washed twice with PBS, fixed with 80% ice-cold acetone at −20 °C for at least 1 h, washed three times with PBS, blocked for 30 min with 3% BSA–PBS, and then treated for 30 min with 2% $H_2O_2$. Cells were incubated with a mouse anti-NP antibody (1:1,000; Millipore) in 3% BSA–PBS for 1 h at room temperature, followed by goat anti-mouse IgG HRP (1:1,000; Southern Biotech) in 3% BSA–PBS for 1 h at room temperature. The plates were developed with Super Aquablue ELISA substrate at 405 nm until virus-only controls reached an OD of 1. The signal from uninfected wells was averaged to represent 100% inhibition. The signal from infected wells without mAb was averaged to represent 0% inhibition. Duplication wells were used to calculate the mean and s.d. of neutralization, and the half-maximal inhibitory concentration ($IC_{50}$) was determined by a sigmoidal dose–response curve. The inhibition ratio (%) was calculated as: ((OD positive control − OD sample)/(OD positive control − OD negative control)) × 100. The final $IC_{50}$ was determined using Prism software (GraphPad). All experiments were performed in duplicate and technically replicated twice.

## H2N2 neutralization assays

Twenty thousand MDCK cells were seeded per well in a 96-well cell culture plate (Corning) and the cells were used the next morning to perform the neutralization assay. Antibody dilutions were prepared starting at 30 µg ml$^{-1}$ with threefold subsequent dilutions in 1X MEM. Each respective dilution was mixed with 10,000 plaque-forming units (PFU) of cold-adapted A/Ann Arbor/6/1960 (H2N2) virus for 1 h at room temperature. After 1 h, cells were washed with PBS and 100 µl of antibody–virus mixture was added onto the cells for 1 h at 37 °C. Next, the antibody–virus mixture was removed and 60 µl of 1X MEM containing TPCK was added to each well. Of each corresponding antibody dilution, 60 µl was also added to each well and the cells were incubated at 33 °C for 3 days. On the third day, a haemagglutination assay was performed using turkey red blood cells to assess the HAU at each antibody dilution.

## In vivo challenge infections

mAb cocktails (Extended Data Fig. 2b) were passively transferred into 6–8-week-old female BALB/c mice (Jackson Laboratories) by

intraperitoneal injection of 0.2, 1 and 5 mg per kg mAb cocktail, which are further detailed in Supplementary Table 1. Negative control mice received 5 mg per kg of the anthrax-specific mAb 003-15D03 as an isotype control. mAbs were administered 2 h before infection for prophylactic treatment and 48 h post-infection for therapeutic treatment. For prophylactic mAb studies with A/Netherlands/602/2009 (Extended Data Fig. 3a), mice were anaesthetized with isoflurane and intranasally challenged with 10 lethal dose 50 ($LD_{50}$) of mouse-adapted A/Netherlands/602/2009 H1N1 virus, with 10 μl of virus administered into each nostril (20 μl total). For therapeutic treatment of A/Netherlands/602/2009 and prophylactic treatment of A/Fort Monmouth/1/1947, mice were anaesthetized with a ketamine–xylazine–water cocktail (0.15 mg ketamine per kg and 0.03 mg per kg xylazine; 100 ml intraperitoneally) and infected with 10 $LD_{50}$ of A/Netherlands/602/2009 or A/Fort Monmouth/1/1947. As a read out, survival and weight loss were monitored 1–2 times daily for 2 weeks. Mice were euthanized upon 25% weight loss or at the end of the experiment (14 days post-challenge). Five mice per condition per experiment with two biological replicates were utilized based on a previously performed power analysis. Data were pooled for analysis.

To determine differences in lung viral load, 5 mg per kg of the antibody cocktails was administered prophylactically as described above. Two hours after mAb administration, mice ($n = 5$ mice per group) were anaesthetized and intranasally challenged with 1 $LD_{50}$ of A/Netherlands/602/2009. Lungs were collected at day 3 and day 6 post-infection, homogenized and viral load was determined via plaque assay. All experiments were done in accordance with the University of Chicago and Icahn School of Medicine at Mount Sinai Institutional Animal Care and Use Committees. Animals studies were not blinded or randomized.

### Plaque assay
For determination of viral load in mouse lung tissues a standard plaque assay was performed. Confluent monolayers of MDCK cells were infected with serial dilutions of homogenized lung tissue ranging from 1:10 to 1:1,000,000 diluted in 1X MEM (1% penicillin–streptomycin antibiotics mix, 1% HEPES, 1% L-glutamine and 1% sodium-bicarbonate (Gibco)) for 1 h at 33 °C, with shaking every 15 min. Afterwards, an overlay containing 2% Oxoid agar (Thermo Fisher), $H_2O$, 2X MEM, DEAE and TPCK-treated trypsin was added to the cells. The plates were incubated at 33 °C for 3 days and then fixed with 3.7% paraformaldehyde overnight at 4 °C. Plaques were visualized by immunostaining. Here, the agar overlay was removed and the plates blocked with 3% milk and PBS. The blocking solution was removed and primary antibody ((H1N1 guinea pig anti-sera (generated in house)) diluted 1:3,000 in 1% milk and PBS was added for 1 h. The plates were washed three times with PBS and secondary antibody (anti-mouse IgG H&L peroxidase-conjugated (Rockland) diluted 1:3,000 in 1% milk and PBS was added for 1 h. The plates were washed three times with PBS and developed by using KPL TrueBlue Peroxidase Substrate (SeraCare).

### Antibody-dependent cellular cytotoxicity reporter assay
A549 cells were maintained in DMEM supplemented with 10% FBS, 10 U ml$^{-1}$ penicillin, and 10 mg ml$^{-1}$ streptomycin) and were plated in 96-well, white-walled plates (Costar) at $2.5 \times 10^5$ cells per ml overnight at 37 °C with 5% $CO_2$. The following day, cells were washed with PBS and infected with A/Netherlands/602/2009 at a multiplicity of infection of 5 in UltraMDCK media (Lonza) for 24 h in the absence of TPCK-treated trypsin. mAbs were serially diluted in assay buffer (RPMI 1640 supplemented with 4% ultra-low IgG FBS; Gibco), starting at 60 μg ml$^{-1}$ and diluted threefold. Cell medium was aspirated and 25 μl of assay buffer and 25 μl of diluted antibody were added to each well. Jurkat cells expressing human FcgRIIIa with a NFAT-driven luciferase reporter gene (Promega) were diluted to $3 \times 10^6$ cells per ml, 25 μl of cells was added to each well and incubated at 37 °C with 5% $CO_2$ for 6 h. Plates were removed from the incubator and placed at room temperature for

15 min. Of the BioGlo luciferase substrate (Promega), 75 μl was added to each well and luminescence was read immediately using a Syngery H1 hybrid multimode microplate reader (Biotek). $EC_{50}$ values were determined using Prism 8 (GraphPad).

### HA–antibody binding footprint mapping
The footprints of three mAbs (FISW84 (PDB: 6HJQ), CR9114 (PDB: 4FQI) and FI6v3 (PDB: 3ZTN)) were mapped onto one HA protomer (A/California/4/2009, PDB: 4M4Y) using UCSF Chimera[60] and Adobe Photoshop. Negative-stain EM maps of HA–Fab complexes were aligned in UCSF Chimera and estimated footprints were mapped onto one HA protomer. Individual protomers of the HA trimer are indicated in different shades of grey.

### Negative-stain EM
Immune complexes were prepared by incubating Fab with HA (A/California/04/2009 with E376K or E376G stabilizing mutations) at greater than 3:1 molar ratio for 2 h at room temperature. Samples were deposited at approximately 10 μg ml$^{-1}$ on glow-discharged, carbon-coated 400 mesh copper grids (Electron Microscopy Sciences) and stained with 2% w/v uranyl formate. Samples were imaged at ×52,000 magnification, 120 kV, on a Tecnai Spirit T12 microscope equipped with an Eagle CCD 4k camera (FEI) or ×62,000 magnification, 200 kV, on a Tecnai T20 microscope equipped with a CMOS 4k camera (TVIPS). Micrographs were collected with Leginon, single particles were processed with Appion, Relion and XQuartz, and footprints were mapped with UCSF Chimera, and figures were made with UCSF Chimera[60–63].

### Cryo-EM
222-1C06 and 045-09 2B05 Fabs were incubated at greater than 3:1 molar ratio with HA (A/California/7/2009, E376K) for 1 h at room temperature. 045-09 2B05 Fab, targeting the lateral patch, was added to the immune complex to induce particle tumbling and increase angular sampling[3]. Using a Thermo Fisher Vitrobot, the immune complex (0.5 mg ml$^{-1}$) incubated with lauryl maltose neopentyl glycol (5 μM, Anatrace) was deposited onto glow-discharged Au 1.2/1.3 300 mesh grids (Electron Microscopy Sciences), blotted for 7 s, and plunge-frozen in liquid ethane. Samples were imaged at ×36,000 nominal magnification on a 200 kV Talos Arctica electron microscope (FEI) with a CETA 4k CMOS camera (FEI, total dose 49.92 e/Å$^2$) and Gatan K2 Summit detector in counting mode. 2,243 micrographs were collected, aligned and CTF-corrected using Leginon, MotionCor2 in Appion, and Patch-CTF in CryoSPARC2, respectively[61,62,64,65]. In CryoSPARC2, particles were picked using apo HA templates, selected through reference-free 2D classification, and reconstructed through 3D classification and refinement. The final map resolved to a global 3.75 Å resolution with C3 symmetry and 44,224 particles. Figures were made in Prism 8 (GraphPad) and UCSF Chimera[60].

### Model building and refinement
A predicted model of 222-1C06 Fab was generated using abYsis (http://www.abysis.org/abysis/) and docked into EM density along with an initial model of CA09 H1 HA + 045-09 2B05 (PDB: 7MEM). The initial model was iteratively refined using COOT and Rosetta[66,67]. The final model was numbered using the H3 and Kabat numbering schemes. The final model and map were evaluated using MolProbity, EMRinger[68,69], Phenix and the PDB validation server. After modelling the immune complex, we segmented the Fab density from HA in the cryo-EM map and mapped the footprint of the 222-1C06 model in the HA density. Cryo-EM data collection and refinement statistics are included in Extended Data Table 4.

### Electron microscopy polyclonal epitope mapping
Human serum samples were heat-inactivated at 55 °C for 30 min before incubating on Capture Select IgG-Fc (ms) Affinity Matrix (Fisher) to

bind IgG at 4 °C for 72 h on a rotator. Samples with IgG bound to resin were centrifuged at 4,000 rpm and supernatant was collected. IgG samples were washed three times with PBS followed by centrifugation to remove supernatant. Samples were buffer exchanged into buffer containing 100 mM Tris, 2 mM EDTA and 10 mM L-cysteine through centrifugation with Amicon filters, then incubated with papain for 4 h at 37 °C, shaking at 80 rpm. The digestion reactions were quenched with 50 mM iodoacetamide, buffer exchanged to TBS and separated by size-exclusion chromatography (SEC) with a Superdex 200 increase 10/300 column (GE Healthcare). Fab and undigested IgG were collected and concentrated, and 500 µg Fab was complexed with 10 µg HA for 18 h at room temperature. Reactions were purified by SEC and immune complexes were collected and concentrated. Negative-stain EM grids were prepared as described above.

## Membrane-bound HA and mAb staining

HEK293T cells were plated into a six-well plate and transfected overnight with 0.2 µg of plasmid and 10 µg ml$^{-1}$ PEI. After 12–16 h, media were replaced with PFHM-II (Gibco) and cells were rested for 3 days. Transfected cells were trypsinized, washed and aliquoted. Cells were stained with 10 µg ml$^{-1}$ of individual mAbs for 30 min. Cells were washed and stained with anti-human IgG Fc-BV421 for 30 min. Cells were washed two times and run on a BD LSRFortessa and collected with BD FACSDiva software. Data were analysed using FlowJo v10.

## Single-cell RNA sequencing and repertoire analysis

cH5/1$^+$ memory B cells (CD19$^+$CD27$^+$HA$^+$) were bulk sorted and partitioned into nanolitre-scale gel bead-in-emulsions (GEMs) to achieve single-cell resolution using the 10x Genomics Chromium Controller and according to the manufacturer's instruction (10x Genomics). The sorted single cells were processed according to 5′ gene expression and B cell immunoglobulin enrichment instruction to prepare the libraries for sequencing. Libraries were sequenced using an Illumina HiSeq 4000 at Northwestern University or an Illumina NextSeq 500 at the University of Chicago. Cellranger Single-Cell Software Suite (version 3.0) was used to perform sample demultiplexing, barcode processing, and single-cell 5′ and V(D)J counting, and Cellranger mkfastq was used to demultiplex raw base call (BCL) files into sample-specific fastq files. Subsequently, GRCh38-1.2.0 and cellranger-vdj-GRCh 38-alta-ensembl-2.0.0 were used as references for the transcriptome and V(D)J assembly, respectively. Cellranger counts and Cellranger vdj package were used to identify gene expression and assemble V(D) J pairs of antibodies.

Single-cell datasets were analysed using Seurat 3 toolkit (Version 3.2.0). We performed conventional pre-process steps for all 20 donors including cell quality control (QC), normalization, identification of highly variable features, data scaling and linear dimensional reduction. More specifically, we only kept cells with more than 200 and less then 2,500 detected genes for the QC step. We also filtered out cells with high mitochondrial gene expression using a 'softThreshold' function in the R package LinQ-View (version 0.99)[70]. We normalized the RNA data using conventional log normalization. We identified 2,000 highly variable genes for each dataset and performed principle component analysis (PCA) in linear dimensional reduction step. We then integrated all 20 single-cell datasets from vaccinated participants to remove batch effects using the Anchor method in Seurat 3. In this analysis, we filtered our dataset and only kept cells with both transcriptome and full length and paired heavy and light chain V(D)J sequences ($n$ = 1,952). From these cells, we identified a group of 'VH1-69/κ' B cells that used the VH1-69 gene and κ-light chain, which is enriched for B cells targeting the BN stalk epitope. We also identified a group of 'anchor epitope'-specific B cells by the following rules: (1) VH locus: VH3-23, VH3-30, VH3-30-3 or VH-3-48; (2) VK locus: VK3-11 or VK3-15; (3) JK locus: JK4 or JK5; (4) K-CDR3 length equal to 10; and (5) a 'NWP' pattern in the K-CDR3 peptide.

## HA conservation modelling

Pan-H1 conservation models are based on consensus strains (listed in Supplementary Table 4) of distinct H1 clades isolated from humans, swine and avian sources, as described in Zhuang et al.[71] and inclusion of the Eurasian swine-like A/swine/Jiangsu/J004/2018 (ref. [23]). To generate the group 1 HA conservation model, we selected one representative sequence for each group 1 HA subtype from FluDB (https://www.fludb.org/; Supplementary Table 5) according to a previous study[72]. A multiple sequence alignment from these HA protein sequences was generated using MUSCLE[73] and the conservation of each residue was quantified using an entropy model[41]. Seasonal H1 conservation models are based on consensus strains of H1N1 viruses (59 strains total) circulating between 1918–1957 and 1976–2019, which was previously described[3]. Amino acid alignments and H3 numbering were performed using Librator[74] and Burke and Smith HA numbering[72].

## Structure prediction

To predict the structures of the investigated Fv fragments (222-1C06, FISW84, 241IgA 2F04 and SFV009 3G01) with A/California/4/2009 E47G HA (PDB: 7MEM), we applied the program RosettaAntibody[67,75,76]. The Fvs were protonated using the Protonate 3D tool[77,78]. Charge neutrality was ensured by utilizing the uniform background plasma approach in AMBER[79,80]. Using the tleap tool of the AmberTools20 (ref. [81]) package, the structure models were soaked in cubic water boxes of TIP3P water molecules with a minimum wall distance of 10 Å to the protein[82]. Parameters for all antibody models derive from the AMBER force field 14SB[83]. The Fvs were carefully equilibrated using a multistep equilibration protocol[84].

## Metadynamics simulations

To enhance the sampling of the conformational space, well-tempered bias-exchange metadynamics[85–87] simulations were performed in GROMACS[88,89] with the PLUMED 2 implementation[90]. We chose metadynamics as it enhances sampling on predefined collective variables. The sampling is accelerated by a history-dependent bias potential, which is constructed in the space of the collective variables[85,86,91]. As collective variables, we used a well-established protocol, boosting a linear combination of sine and cosine of the ψ torsion angles of all six CDR loops calculated with functions MATHEVAL and COMBINE implemented in PLUMED 2 (ref. [90]). As discussed previously, the ψ torsion angle captures conformational transitions comprehensively[92]. The underlying method presented in this paper has been validated in various studies against a large number of experimental results[93]. The simulations were performed at 300 K in an NpT ensemble using the GPU implementation of the pmemd module[94] to be as close to the experimental conditions as possible and to obtain the correct density distributions of both protein and water. We used a Gaussian height of 10.0 kJ mol$^{-1}$ and a width of 0.3 rad. Gaussian deposition occurred every 1,000 steps and a biasfactor of 10 was used. 500 ns of bias-exchange metadynamics simulations were performed for the prepared Fv structures. The resulting trajectories were aligned to the whole Fv and clustered with the program cpptraj[80,95] using the average linkage hierarchical clustering algorithm with a RMSD cut-off criterion of 1.2 Å resulting in a large number of clusters. The cluster representatives for the antibody fragments were equilibrated and simulated for 100 ns using the AMBER 20 (ref. [81]) simulation package.

## Molecular dynamics simulations

Molecular dynamics simulations were performed in an NpT ensemble using the pmemd.cuda module of AMBER 20 (ref. [80]). Bonds involving hydrogen atoms were restrained with the SHAKE algorithm[96], allowing a time step of 2.0 fs. Atmospheric pressure (1 bar) of the system was set by weak coupling to an external bath using the Berendsen algorithm[97]. The Langevin thermostat[98] was used to maintain the temperature during simulations at 300 K.

With the obtained trajectories, we performed a time-lagged independent component analysis (tICA) using the Python library PyEMMA 2, using a lag time of 10 ns. tICA was applied to identify the slowest movements of the investigated Fab fragments and consequently to obtain a kinetic discretization of the sampled conformational space[99]. On the basis of the tICA conformational spaces, thermodynamics and kinetics were calculated with a Markov-state model[100] of all six CDR loops by using PyEMMA 2. The resulting kinetically dominant ensemble in solution was further used to predict the interactions of H1 with the Fvs. To model the complex and to predict interactions in the binding interface, we used the crystal structure of the full-length influenza HA (PDB: 7MEM) as template structure. In addition, the obtained complex structure was further minimized and equilibrated.

## Statistical analysis

All statistical analyses were performed using Prism software (GraphPad versions 8 and 9) or R. Sample sizes ($n$) for the number of mAbs tested are indicated in corresponding figures or in the centre of pie graphs. The number of biological repeats for experiments and specific tests for statistical significance used are indicated in the corresponding figure legends. $P$ values less than or equal to 0.05 were considered significant: $*P \leq 0.05$, $**P \leq 0.01$, $***P \leq 0.001$ and $****P < 0.0001$.

## Reporting summary

Further information on research design is available in the Nature Research Reporting Summary linked to this paper.

## Data availability

Repertoire data generated from single-cell RNA sequencing data were deposited at Mendeley Data (https://data.mendeley.com/datasets/jzsx489pmk/1). Accession numbers for all other anchor-targeting mAbs are included in Supplementary Table 6. Electron microscopy maps were deposited to the Electron Microscopy Data Bank under accession IDs: EMD-25634–EMD-25646. The cryo-EM map and model of anchor and lateral patch Fabs binding H1 HA were deposited to the RCSB database with accession numbers EMD-25655/PDB 7T3D. All next-generation sequencing data for 045-09 2B06 deep mutational scanning and for the H1N1 mutational scanning can be found on the Sequence Read Archive under BioProject accession number PRJNA309339. The following Protein Data Bank accession numbers were downloaded and included in the paper: 6HJQ, 3SDY, 4NM8, 4M4Y, 4WE4, 4JTV, 4FQI, 3ZTN and 7MEM. Sera from the vaccine cohorts are unique to this study and are not publicly available. All source data are included with the paper. All other material is available on reasonable request to the corresponding authors. Source data are provided with this paper.

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

**Acknowledgements** We thank all of the volunteers who participated in this study; S. Andrews, R. Ahmed, J. Wrammert and K. Neu for initiating studies on the 2009 MIV, 2010 TIV and 2014 QIV cohorts; C. Mariottini, J. Feser, D. Stadlbauer and A.-K. Palm for their help on the cHA vaccine trial; I. Wilson for fruitful discussion and feedback on experimental design; B. Anderson and H. Turner for assistance with electron microscopy; and the teams at PATH, GSK, Cincinnati Children's Hospital Medical Center, and Duke University for their work on the cHA vaccine trial (NCT03300050). This project was funded in part by the National Institute of Allergy and Infectious Diseases (NIAID; National Institutes of Health grant numbers K99AI159136 (to J.J.G.), U19AI082724 (to P.C.W.), U19AI109946 (to P.C.W.), U19AI057266 (to P.C.W.), P01AI097092 (to P.P.), R01AI145870-01 (to P.P.), R21AI146529 (to L.C.) and T32AI007244-36 (to J.H.), the NIAID Centers of Excellence for Influenza Research and Surveillance (CEIRS) grant number HHSN272201400005C (to P.C.W.), HHSN272201400008C (to L.C., F.K., A.G.-S. and P.P.), and the NIAID Centers of Excellence for Influenza Research and Response (CEIRR) grant number 75N93019R00028 (to P.C.W., F.K., A.G.-S. and P.P.). This work was also partially supported by the NIAID Collaborative Influenza Vaccine Innovation Centers (CIVIC; 75N93019C00051, to F.K., A.G.-S., P.P., A.B.W. and P.C.W.). The cHA vaccine trial and evaluation of immunity thereafter was funded in part by the Bill and Melinda Gates Foundation (OPP1084518). The findings and conclusions contained within are those of the authors and do not necessarily reflect positions or policies of the Gates Foundation. This study was also funded by the Austrian Science Fund grant number P34518 (to M.L.F.-Q. and K.R.L.).

**Author contributions** J.J.G., J.H., A.B.W. and P.C.W. conceptualized the study. J.J.G., J.H., H.A.U., L.L., L.Y.-L.L., A.W.F., M.L.F.-Q., C.H., C.T.S., M.M., G.O., F.A., L.G., N.-Y.Z., S.S., M.H., J.D.B. and L.C. developed the methodology. J.J.G., J.H., H.A.U., L.L., L.Y.-L.L., A.W.F., M.L.F.-Q., C.H., C.T.S., M.M., G.O., V.R., F.A., O.S., L.G., H.L.D., S.T.R., A.T.d.l.P., M.E.T., D.J.B., S.C. and L.C. conducted the investigation. J.J.G., J.H., H.A.U., L.L., M.L.F.-Q., M.M., L.G. and S.T.R. performed visualization. J.J.G., J.H., A.G.-S., P.P., F.K., L.C., A.B.W. and P.C.W. acquired funding. Project administration was done by J.J.G., J.H., C.H., A.G.-S., K.R.L., R.N., P.P., F.K., A.B.W. and P.C.W. J.J.G., J.H., A.B.W. and P.C.W. supervised the study. J.J.G. and J.H. wrote the original draft of the manuscript. J.J.G., J.H., H.A.U., L.L., L.Y.-L.L., C.H., C.T.S., M.M., G.O., A.W.F., O.S., L.G., H.L.D., N.-Y.Z., S.T.R., A.T.d.l.P., M.E.T., D.J.B., S.C., S.S., M.H., A.G.-S., R.N., P.P., J.D.B., F.K., L.C., A.B.W. and P.C.W. reviewed and edited the manuscript.

**Competing interests** The Icahn School of Medicine at Mount Sinai has patents (10736956, 10583188, 10137189, 10131695, 9968670, 9371366) and has submitted patent applications (10736956, 10583188, 10137189, 10131695, 9968670, 9371366) on universal influenza virus vaccines naming A.G.-S., R.N., P.P. and F.K. as inventors. The University of Chicago has submitted patent applications (632292804) on anti-anchor mAbs naming J.J.G. and P.C.W. as inventors. The A.G.-S. laboratory has received research support from Pfizer, Senhwa Biosciences, Kenall Manufacturing, Avimex, Johnson & Johnson, Dynavax, 7Hills Pharma, Pharmamar, ImmunityBio, Accurius, Nanocomposix, Hexamer, N-fold LLC and Merck, outside of the reported work. A.G.-S. has consulting agreements for the following companies involving cash and/or stock: Vivaldi Biosciences, Contrafect, 7Hills Pharma, Avimex, Vaxalto, Pagoda, Accurius, Esperovax, Farmak, Applied Biological Laboratories and Pfizer, outside of the reported work.

**Additional information**
**Correspondence and requests for materials** should be addressed to Jenna J. Guthmiller, Andrew B. Ward or Patrick C. Wilson.

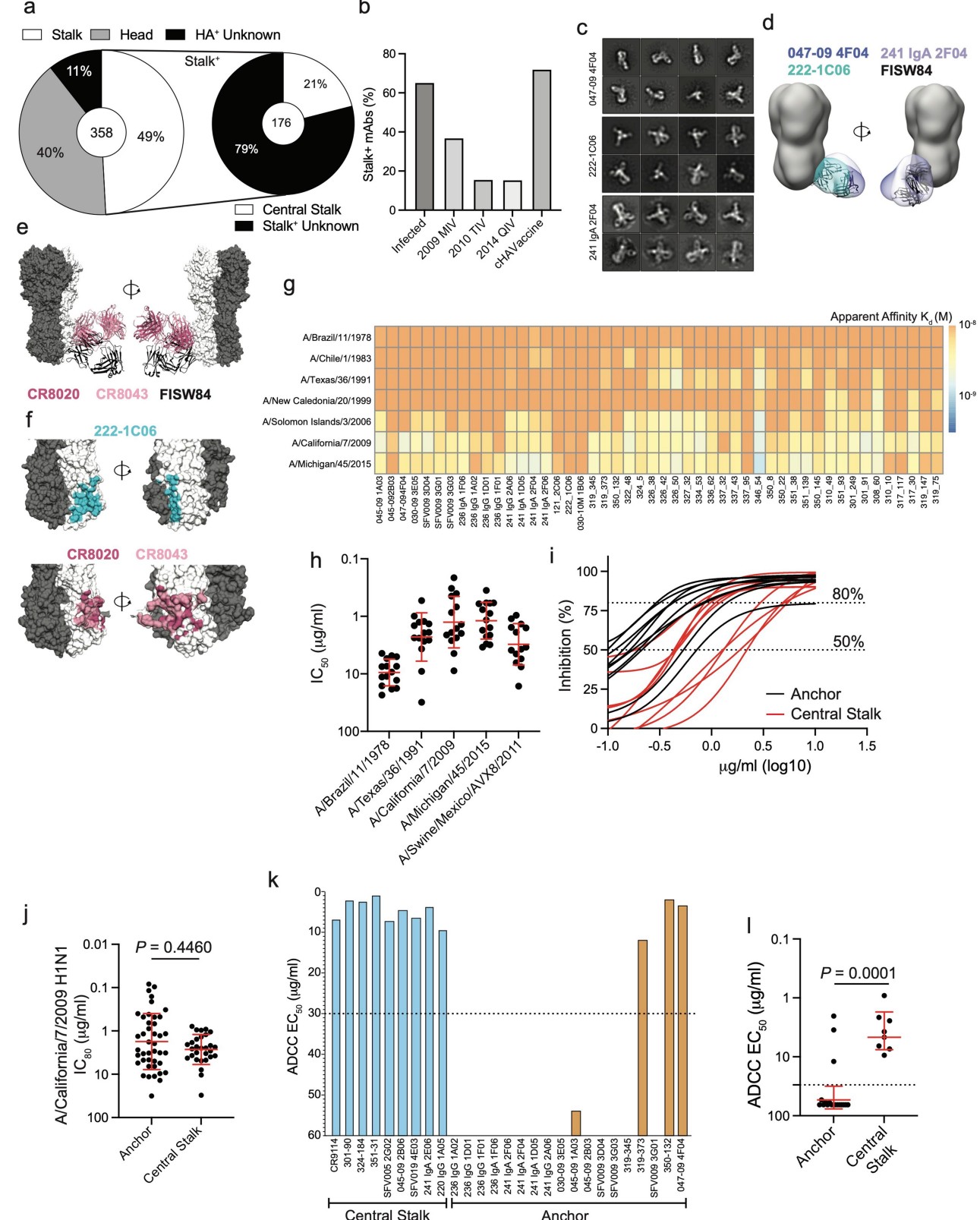

**Extended Data Fig. 1** | See next page for caption.

**Extended Data Fig. 1 | Binding and neutralization features of anchor epitope-binding mAbs. Related to Fig. 1. a**, Proportion of HA[+] mAbs binding to distinct HA domains (left) and proportion of stalk-binding mAbs binding the CS domain (right). Number in the center of the pie graphs represent the number of mAbs tested. **b**, Proportion of mAbs per cohort that bind the HA stalk domain. **c**, Negative stain 2D class averages of 047-09 4F04, 241 IgA 2F04, and 222-1C06 binding to H1 (A/California/7/2009 E376K HA). Imaging of 047-09 4F04 was performed at ×52,000 normal magnification and of 222-1C06 and 241 IgA 2F04 at ×62,000 normal magnification. **d**, Overlay of 047-09 4F04, 241 IgA 2F04, 222-1C06, and FISW84 (PDB:6HJQ) Fabs binding the anchor epitope of A/California/4/2009 HA. **e**, Overlay of CR8020 (PDB:3SDY), CR8043 (PDB:4NM8), and FISW84 (PDB:6HJQ) modeled on A/California/7/2009 E376G (PDB:4M4Y). **f**, Footprints of anchor mAb 222-1C06 on H1 (top; PDB: 4M4Y) and CR8020 and CR8043 on H3 (bottom; PDB:4WE4). **g**, Heatmap of apparent affinity ($K_d$; M) of anchor-targeting mAbs binding to historical and recent H1N1 viruses. **h**, Neutralization potency of anchor-binding mAbs (n = 15) against H1-expressing viruses. **i**, Representative microneutralization curves of anchor- (n = 42) and CS-binding (n = 29) mAbs against A/California/7/2009. **j**, $IC_{80}$ of anchor- and CS-binding mAbs against A/California/7/2009. **k**, ADCC activity of mAbs targeting the CS and anchor epitopes. Dashed line represents the limit of detection (L.O.D). **l**, ADCC potency of mAbs targeting the anchor (n = 18 mAbs) and CS (n = 8 mAbs) epitopes. Data in **h**, **j**, and **l** are represented as mean ± S.D. Data in **j** and **l** were analyzed using a two-tailed unpaired non-parametric Mann-Whitney test.

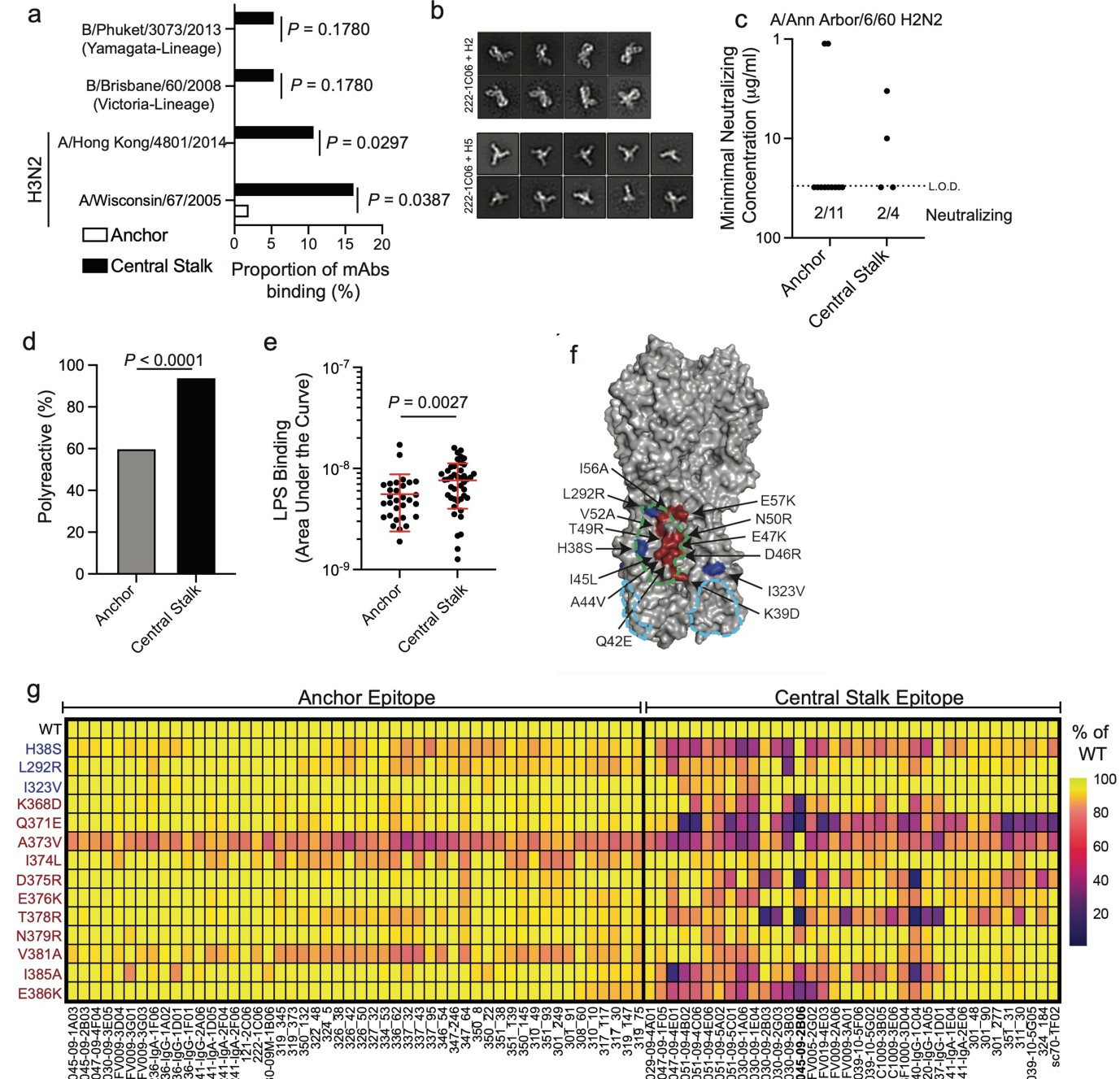

**Extended Data Fig. 2 | Anchor-targeting mAb binding to influenza subtype, viral mutants, and polyreactivity antigen panel. Related to Fig. 1.**
**a**, Proportion of anchor- (n = 50 mAbs) and CS-targeting mAbs (n = 37) binding influenza B viruses and H3N2 viruses. **b**, Negative stain 2D class averages (×62,000 normal magnification) of 222-1C06 binding to H2 (A/Singapore/1/1957), and H5 (A/Indonesia/5/2005). **c**, H2N2 neutralizing data of anchor- (n = 11 mAbs) and CS-binding mAbs (n = 4) represented as minimum neutralizing concentration. The limit of detection (L.O.D.) is 30 mg/ml. **d**, Proportion of mAbs targeting the anchor (n = 50 mAbs) or CS (n = 50 mAbs) epitope that are polyreactive. **e**, LPS binding strength, represented as area under the curve (AUC), of polyreactive mAbs targeting the anchor (n = 30

mAbs) and central stalk (n = 43 mAbs) epitopes. Data are mean ± S.D. **f**, **g**, Anchor- and CS epitope-binding mAbs were tested for binding to A/California/7/2009 HA with naturally occurring and experimentally determined mutations induced by 045-09 2B06, a CS-binding mAb. **f**, Location of mutations modeled on A/California/4/2009 HA (PDB: 4JTV). Residues in blue are located on HA1 and residues in red are located on HA2. Outlines represent binding footprints of 047-09 4F04 (sky blue) and CR9114 (green). **g**, Heatmap of mAb binding to WT and mutant HAs shown as the proportion of signal relative to mAb binding to the WT HA. Data in **a** and **d** were analyzed using Fisher's Exact tests. Data in **e** were analyzed using a two-tailed unpaired non-parametric Mann-Whitney test.

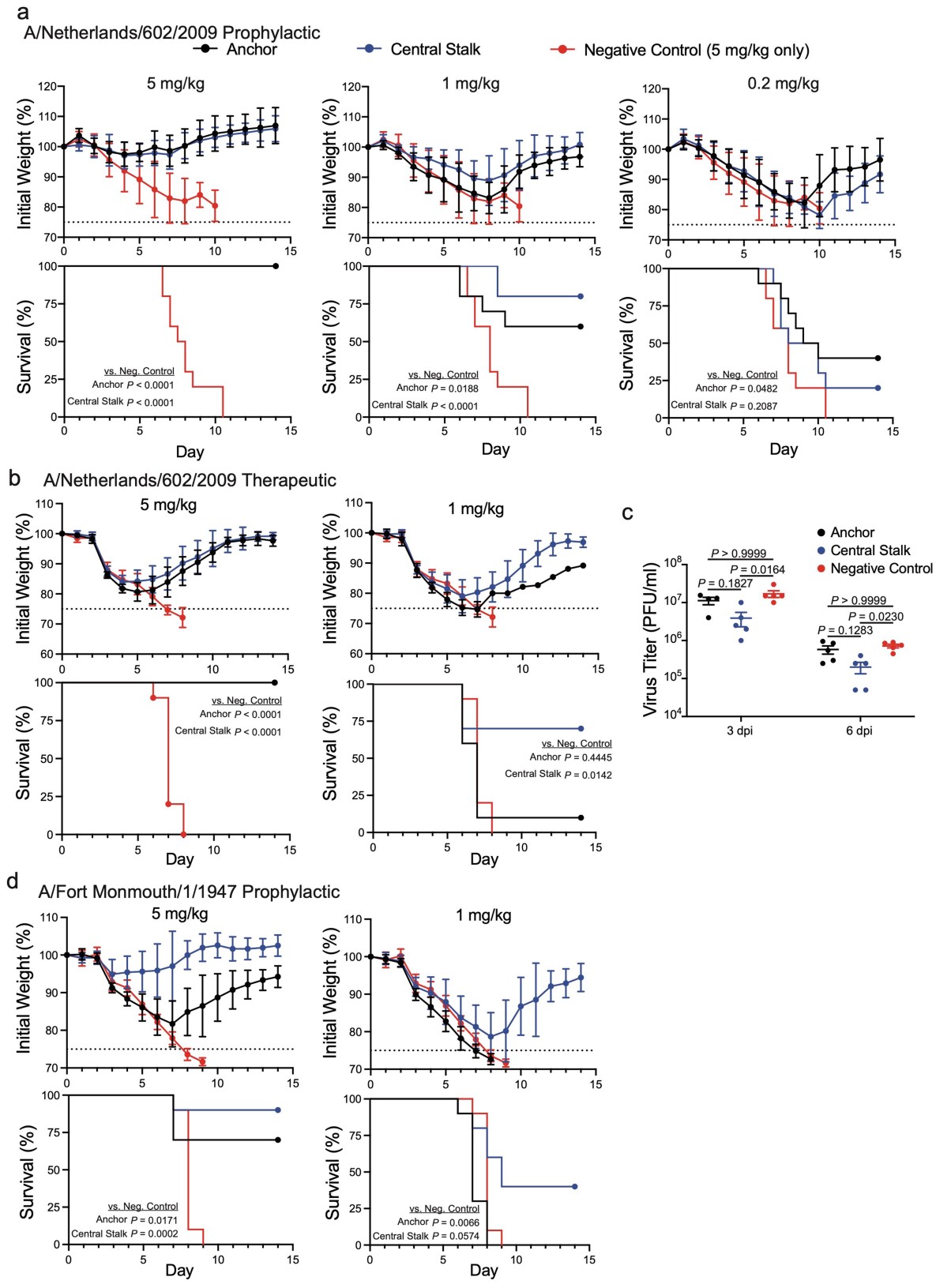

**Extended Data Fig. 3** | See next page for caption.

**Extended Data Fig. 3 | Anchor epitope-targeting mAbs are potently protective *in vivo* and lack ADCC activity. Related to Fig. 1. a**, **b**, Mice were prophylactically (2 h prior to infection; **a**) or therapeutically (48 h after infection; **b**) administered i.p. a cocktail of mAbs (n = 5 mAbs/cocktail) against the anchor- or CS-, or an anthrax-specific antibody. Mice were infected with 10 LD$_{50}$ of A/Netherlands/602/2009 H1N1. Weight loss (top) and survival (bottom) of mice in each treatment group. **c**, Lung viral titers of mice in each prophylactic treatment group infected with 1 LD$_{50}$ of A/Netherlands/602/2009. dpi, days post infection. **d**, Mice were prophylactically (2 h prior to infection) administered i.p. a cocktail of mAbs (n = 5 mAbs/cocktail) against the anchor- or CS-, or an anthrax-specific antibody. Mice were infected with 10 LD$_{50}$ of A/Fort Monmouth/1/1947 H1N1. Weight loss (top) and survival (bottom) of mice in each treatment group. For **a**, **b**, and **d**, 10 mice per treatment group were used and data are pooled from two independent experiments. For **c**, 5 mice per treatment group and timepoint were used except for anchor cocktail group at dpi 3 only 4 mice were used. Data in **a**, **b**, and **d** are represented as mean ± S.D and data in **c** are represented by mean ± S.E.M. Kaplan-Meier curves in **a**, **b**, **d** were analyzed using a Log-rank Mantel-Cox test, and data in **c** were analyzed using multiple two-tailed unpaired non-parametric Kruskal-Wallis tests.

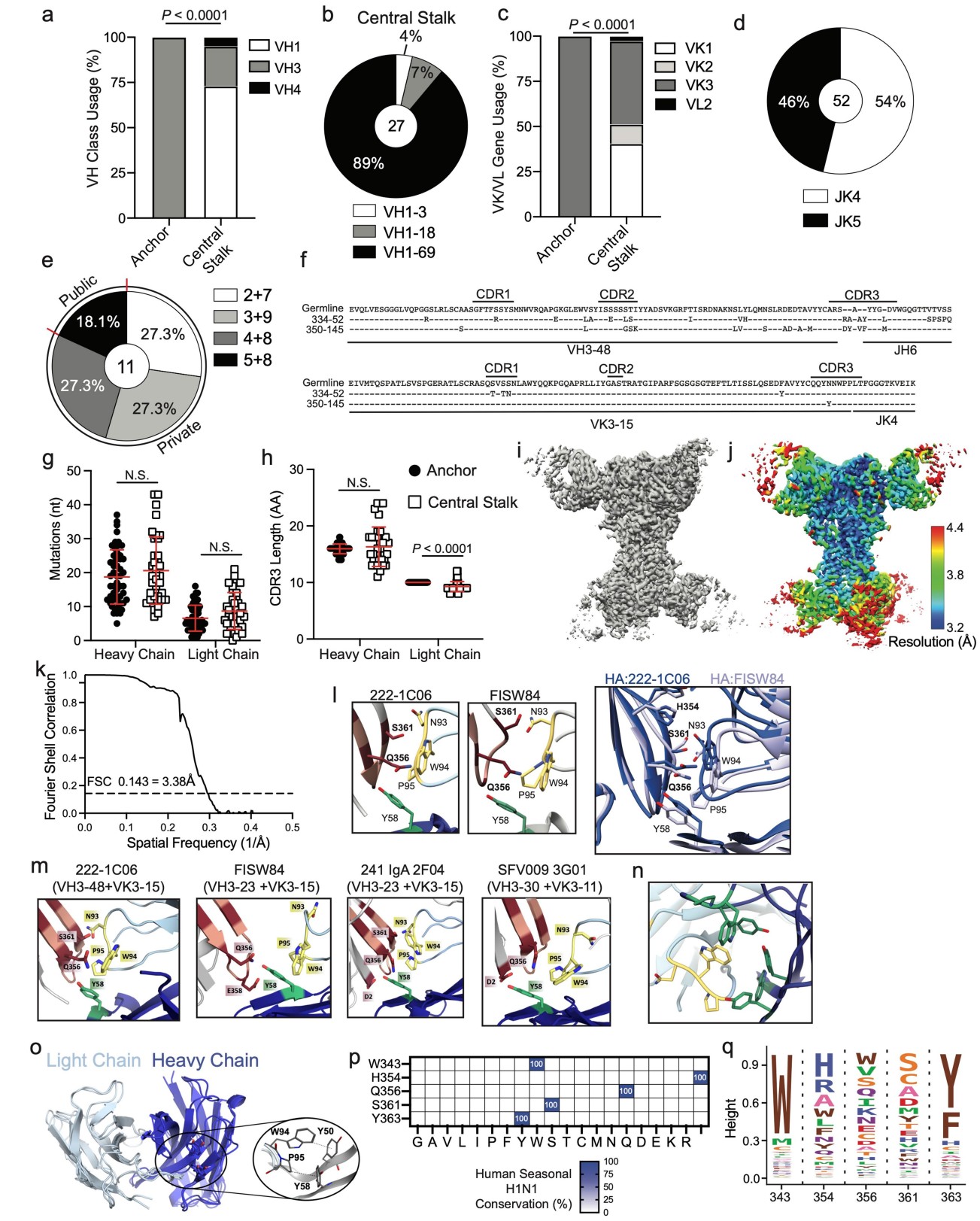

**Extended Data Fig. 4** | See next page for caption.

**Extended Data Fig. 4 | Additional repertoire and structural features of mAbs binding the anchor epitope. Related to Fig. 2. a**, VH locus usage by anchor- (n = 52 mAbs) and CS-binding mAbs (n = 37 mAbs). **b**, VH1 gene usage of mAbs targeting the CS epitope. **c**, VK locus usage by anchor- (n = 52 mAbs) and CS-binding mAbs (n = 37 mAbs). **d**, JK gene usage by anchor epitope-binding mAbs. **e**, Clonal expansions of anchor epitope-targeting mAbs. Numbers indicate heavy and light chain parings, which are described in Extended Data Table 2. **f**, Heavy and light chain sequences of the public clone. **g**, **h**, Mutations (**g**) and CDR3 amino acid (AA) lengths (**h**) of heavy and light chains of mAbs binding the anchor (n = 52 mabs) or CS (n = 37 mAbs) epitopes. Data are mean ± S.D. **i**, Cryo-EM map of 222-1C06 binding to A/California/7/2009 E376K HA. **j**, **k**, Local resolution (**j**) and Fourier Shell Correlation (**k**) of 222-1C06 binding to HA. **l**, Aromatic pockets of 222-1C06 binding A/California/7/2009 E376K and FISW84 binding to A/duck/Alberta/35/1976 (PDB:6HJQ; top) and overlay of epitope:paratope interaction (bottom). **m**, MD simulations demonstrating the K-CDR3 NWP and H-CDR2 Y58 motifs of 222-1C06, FISW84, 241 IgA 2F04, and SFV009 3G01 binding to HA A/California/7/2009 HA. For left-hand panels in **l** and all panels in **m**, HA epitope contact residues (maroon) and heavy chain (green) and light chain (yellow) antibody contact residues of anchor mAb paratopes. Peach highlighted amino acids represent the fusion peptide of HA2. **n**, Fab-Fab interactions of the aromatic pocket of 222-1C06. **o**, MD simulation of the paratope flexibility of 222-1C06, highlighting the p-stacking of H-CDR2 and K-CDR3. **p**, Conservation of side-chain contacts of 222-1C06 across seasonal human H1N1 viruses circulating between 1918-2019. **q**, Deep mutational scanning of the side-chain contacts of 222-1C06. Data in **a** and **c** were analyzed using a Chi-square test, and data in **g**, **h** were analyzed by two-tailed unpaired non-parametric Mann-Whitney tests.

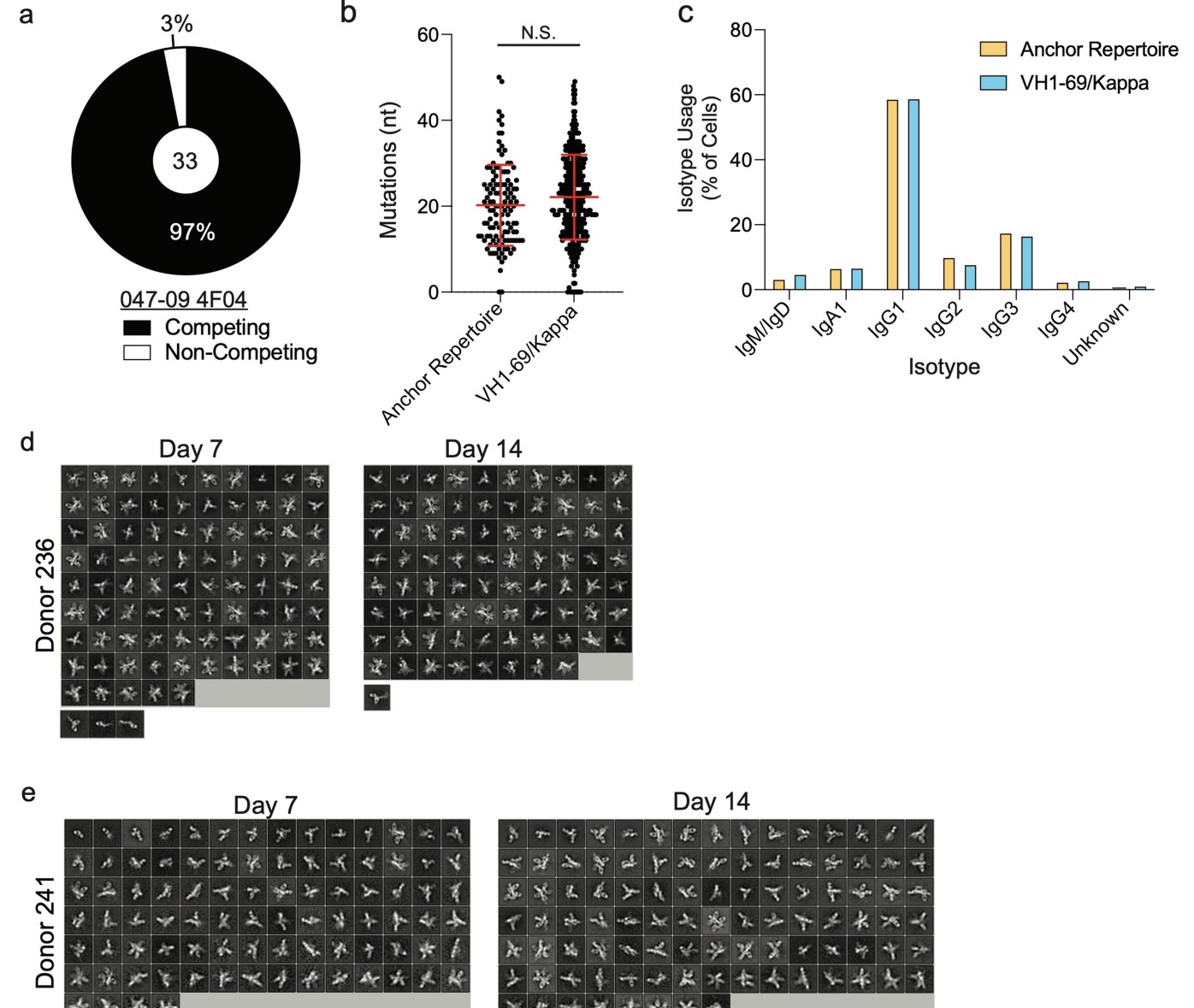

**Extended Data Fig. 5 | Features of anchor-targeting MBCs and EMPEM 2D classes. Related to Fig. 3. a**, 33 mAbs with anchor epitope-binding mAb repertoire features were generated and tested for competing for binding with 047-09 4F04. **b**, **c**, Number of heavy chain mutations (**b**) and isotype usage (**c**) of B cells with repertoire features of anchor-binding mAbs (n = 119 cells) or utilize VH1-69/kappa (n = 365 cells). **d**, **e**, 2D class averages of pAbs from donors 236 (**d**) and 241 (**e**) at days 7 and 14 post immunization binding to A/Michigan/45/2015 HA (×62,000 normal magnification). The last row of 2D classes in **d** is HA monomer complexes processed independently from trimer complexes. Data in **b** are represented as mean ± S.D. Data in **b** were analyzed using a two-tailed paired non-parametric Wilcoxon matched-pairs signed rank test.

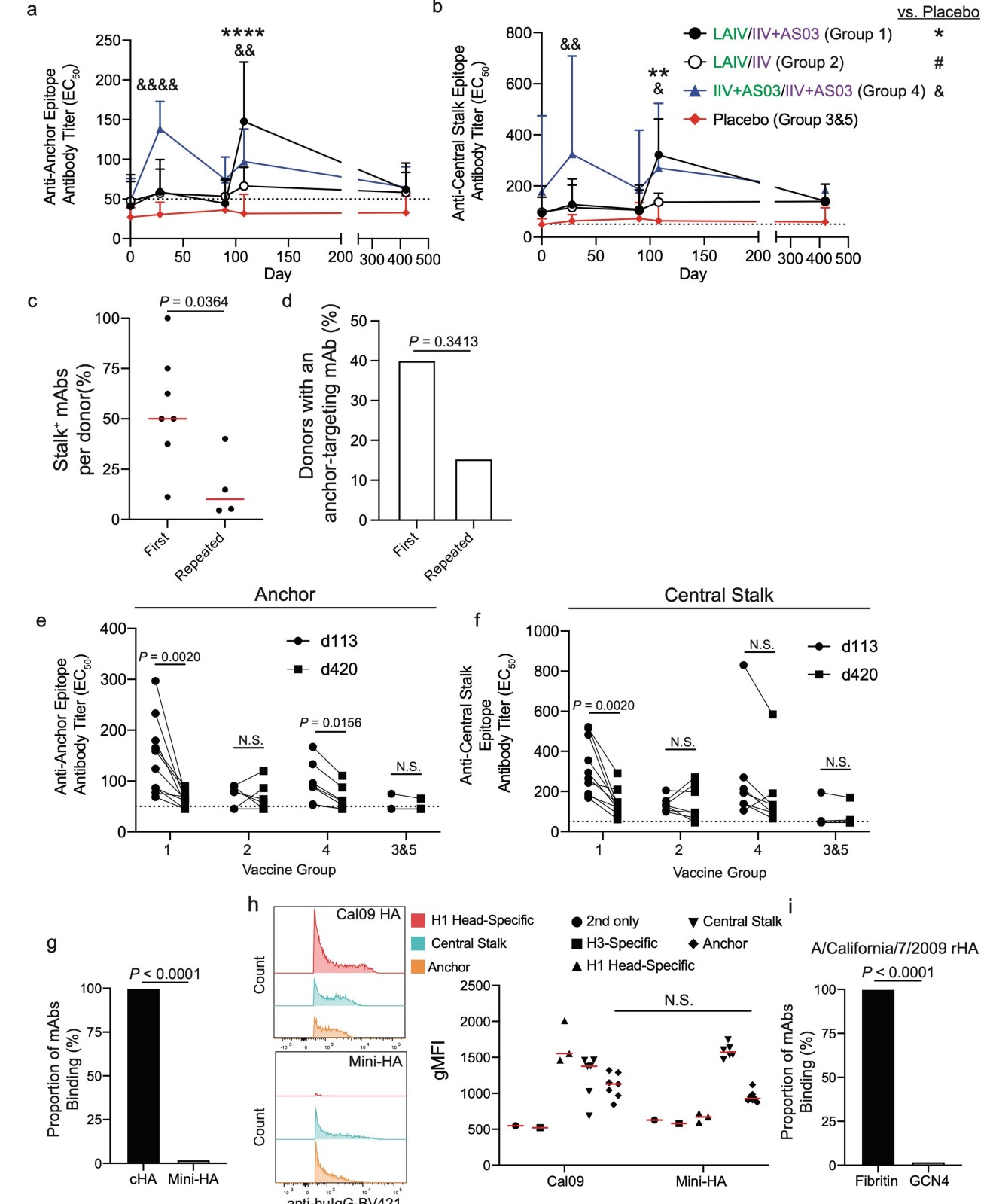

**Extended Data Fig. 6** | See next page for caption.

**Extended Data Fig. 6 | Serum antibody kinetics of anchor- and CS-epitope binding antibodies after cHA vaccination and mAb binding to recombinant HAs. Related to Fig. 4. a**, **b**, EC50s of serum antibodies competing for binding with O47-09 4F04 for binding to the anchor epitope (**a**) and CR9114 for binding to the CS epitope (**b**). **a**, **b**, Kinetics of serum antibody responses against the anchor (**a**) and CS (**b**) epitopes. Data are mean + S.D. **c**, **d**, Proportion of stalk$^+$ mAbs per donor (**c**) or proportion of donors with an isolated anchor mAb (**d**) upon first exposure to the pH1N1 virus (2009 MIV cohort) relative to donors who have repeatedly been exposed to pH1N1 (2010 TIV and 2014 QIV). Data in **c** are mean ± S.D. Data in **c** includes only donors with an isolated anti-stalk mAb, whereas **d** includes all donors. **e**, **f**, Antibody titers (EC$_{50}$) of serum antibodies collected on day 113 and day 420 against the anchor epitope (**e**) and the CS epitope (**f**). Lines connect titers from the same donor and each pair of symbols represents one donor. **g**, Proportion of anchor epitope-binding mAbs binding to cHA (cH6/1) or mini-HA (n = 50). **h**, Representative flow cytometry plots of mAbs binding to A/California/7/2009 Cal09 HA and mini-HA (left) and geometric mean fluorescence intensity (gMFI) of mAbs binding to Cal09 and mini-HA (right). Data represent the mean ± S.D. and each symbol represents an individual mAb. **i**, Proportion of anchor epitope-targeting mAbs binding to A/California/7/2009 recombinant HA with a GCN4 or fibritin trimerization domain (n = 50). For data in **a**, **b**, **e**, and **f**, Group 1 n = 10 participants, group 2 n = 7 participants, group 4 n = 7 participants, group 3&5 n = 6. For data in **c**, first exposure n = 7 donors and repeated exposure n = 4 donors. For data in **d**, first exposure n = 10 donors and repeated exposure n = 13 donors. Data in **a**, **b** were analyzed using a two-tailed two-way ANOVA testing for simple effects within rows, data in **c** and **h** were analyzed using a two-tailed unpaired non-parametric Mann-Whitney test, data in **d**, **g**, and **i** were analyzed using Fisher's Exact test, and data in **c**, **d** were analyzed using a two-tailed paired non-parametric Wilcoxon matched-pairs signed rank test. See also Supplementary Fig. 1 for gating strategy for panel **h**.

**Extended Data Table 1 | Donor information and demographics. Related to Fig. 1**

| Donor | Vaccine/Infection | Age | Sex |
|---|---|---|---|
| 029-09 | 2009 MIV | 24 | M |
| 030-09 | 2009 MIV | 31 | F |
| 045-09 | 2009 MIV | 24 | M |
| 047-09 | 2009 MIV | 22 | M |
| 051-09 | 2009 MIV | 42 | M |
| SFV005 | 2009 MIV | 60 | F |
| SFV009 | 2009 MIV | 31 | F |
| SFV015 | 2009 MIV | 48 | F |
| SFV018 | 2009 MIV | 58 | F |
| SFV019 | 2009 MIV | 48 | F |
| SFV020 | 2009 MIV | 64 | F |
| sc70 | 2009 pH1N1 Infection | 38 | F |
| sc1009 | 2009 pH1N1 Infection | 21 | M |
| SF1000 | 2009 pH1N1 Infection | 37 | M |
| 008-10 | 2010 TIV | 26 | F |
| 011-10 | 2010 TIV | 30 | M |
| 014-10 | 2010 TIV | 27 | M |
| 017-10 | 2010 TIV | 24 | M |
| 019-10 | 2010 TIV | 22 | F |
| 028-10 | 2010 TIV | 38 | M |
| 034-10 | 2010 TIV | 39 | F |
| 039-10 | 2010 TIV | 25 | M |
| 051-10 | 2010 TIV | 43 | M |
| 220-14 | 2014 QIV | 24 | F |
| 221-14 | 2014 QIV | 34 | F |
| 236-14 | 2014 QIV | 32 | F |
| 237-14 | 2014 QIV | 32 | F |
| 240-14 | 2014 QIV | 28 | M |
| 241-14 | 2014 QIV | 29 | F |
| 121/324 (d8/d113) | cHA – Group 4 | 24 | F |
| 222/310 (d92/d113) | cHA – Group 1 | 25 | F |
| 301 | cHA – Group 4 | 34 | F |
| 308 | cHA – Group 1 | 29 | F |
| 311 | cHA – Group 4 | 28 | M |
| 317 | cHA – Group 1 | 37 | M |
| 319 | cHA – Group 1 | 31 | F |
| 322 | cHA – Group 4 | 20 | F |
| 326 | cHA – Group 2 | 27 | M |
| 327 | cHA – Group 1 | 22 | F |
| 328 | cHA – Group 2 | 40 | M |
| 334 | cHA – Group 1 | 33 | M |
| 336 | cHA – Group 1 | 36 | F |
| 337 | cHA – Group 1 | 27 | F |
| 342 | cHA – Group 2 | 22 | M |
| 343 | cHA – Group 4 | 27 | M |
| 346 | cHA – Group 2 | 20 | F |
| 347 | cHA – Group 1 | 31 | F |
| 350 | cHA – Group 1 | 35 | F |
| 351 | cHA – Group 4 | 27 | F |

2009 MIV – 2009 pandemic H1N1 monovalent inactivated influenza vaccine; pH1N1 – 2009 pandemic H1N1 virus; 2010 TIV – 2010-2011 trivalent inactivated influenza vaccine; 2014 QIV – 2014–2015 quadrivant inactivated influenza vaccine; cHA – Group 1, primed with cH8/1 LAIV and boosted with cH5/1 IIV+AS03; cHA – Group 2, primed with cH8/1 LAIV and boosted with cH5/1 IIV; cHA – Group 4, primed with cH8/1 IIV+AS03 and boosted with cH5/1 IIV+AS03. LAIV – live-attenuated influenza vaccine; IIV – inactivated influenza vaccine; AS03 – Adjuvant System 03.

**Extended Data Table 2 | Anchor epitope-binding mAb information. Related to Fig. 1 and 2**

| mAb | Cellular Source | VH | DH | JH | HC Clone # | VK | JK | KC Clone # |
|---|---|---|---|---|---|---|---|---|
| 030-09 3E05 | Plasmablast | 3-23 | 3-22 | 4 | Non-clonal | 3-15 | 5 | 9 |
| 030-09M 1B06 | MBC | 3-23 | 3-10 | 4 | Non-clonal | 3-11 | 5 | 7 |
| 045-09 1A03 | Plasmablast | 3-23 | 6-13 | 4 | Non-clonal | 3-11 | 4 | 6 |
| 045-09 2B03 | Plasmablast | 3-23 | 5-12 | 4 | Non-clonal | 3-11 | 5 | 7 |
| 047-09 4F04 | Plasmablast | 3-48 | 6-19 | 4 | Non-clonal | 3-15 | 4 | 8 |
| SFV009 3D04 | Plasmablast | 3-23 | 2-2 | 4 | Non-clonal | 3-15 | 4 | 8 |
| SFV009 3G01 | Plasmablast | 3-30 | 3-16 | 4 | Non-clonal | 3-11 | 5 | 7 |
| SFV009 3G03 | Plasmablast | 3-23 | 3-15-3 | 5 | Non-clonal | 3-11 | 5 | 7 |
| 236 IgG 1A02 | Plasmablast | 3-23 | 2-2 | 4 | 2 | 3-11 | 5 | 7 |
| 236 IgG 1D01 | Plasmablast | 3-23 | 6-19 | 4 | Non-clonal | 3-11 | 5 | 7 |
| 236 IgG 1F01 | Plasmablast | 3-23 | 2-2 | 4 | 2 | 3-11 | 5 | 7 |
| 236 IgA 1F06 | Plasmablast | 3-23 | 2-2 | 4 | 2 | 3-11 | 5 | 7 |
| 241 IgG 2A06 | Plasmablast | 3-23 | 6-19 | 5 | Non-clonal | 3-15 | 4 | 8 |
| 241 IgA 1D05 | Plasmablast | 3-23 | 5-18 | 4 | 3 | 3-15 | 5 | 9 |
| 241 IgA 2F04 | Plasmablast | 3-23 | 2-21 | 4 | 3 | 3-15 | 5 | 9 |
| 241 IgA 2F06 | Plasmablast | 3-23 | 3-9 | 4 | 3 | 3-15 | 5 | 9 |
| 121 2C06 | Plasmablast | 3-23 | 2-15 | 4 | Non-clonal | 3-15 | 4 | 8 |
| 222 1C06 | Plasmablast | 3-48 | 2-15-2 | 4 | Non-clonal | 3-15 | 4 | 8 |
| 301_91 | MBC | 3-23 | 3-9 | 5 | Non-clonal | 3-11 | 5 | 7 |
| 301_249 | MBC | 3-48 | 5-24 | 3 | Non-clonal | 3-11 | 4 | Non-clonal |
| 301_275 | MBC | 3-23 | 2-21 | 4 | 1 | 3-15 | 4 | 8 |
| 308_60 | MBC | 3-23 | 1-26 | 4 | Non-clonal | 3-11 | 4 | 8 |
| 310_10 | MBC | 3-23 | 1-14 | 4 | Non-clonal | 3-15 | 4 | 8 |
| 310_49 | MBC | 3-23 | 3-22 | 4 | Non-clonal | 3-15 | 5 | 9 |
| 317_30 | MBC | 3-23 | 6-13 | 4 | Non-clonal | 3-15 | 4 | 8 |
| 317_117 | MBC | 3-23 | 2-2 | 4 | Non-clonal | 3-15 | 5 | 9 |
| 319_75 | MBC | 3-23 | 3-22 | 3 | Non-clonal | 3-11 | 5 | 7 |
| 319_147 | MBC | 3-48 | 3-15-3 | 6 | Non-clonal | 3-15 | 4 | 8 |
| 319_345 | MBC | 3-23 | 3-9 | 4 | 4 | 3-15 | 4 | 8 |
| 319_373 | MBC | 3-23 | 6-19 | 4 | 4 | 3-15 | 4 | 8 |
| 322_48 | MBC | 3-23 | 1-26 | 4 | 1 | 3-15 | 5 | 9 |
| 324_5 | MBC | 3-23 | 2-21 | 3 | Non-clonal | 3-15 | 5 | 9 |
| 326_38 | MBC | 3-23 | 1-26 | 4 | Non-clonal | 3-11 | 4 | 6 |
| 326_42 | MBC | 3-23 | 3-16 | 5 | Non-clonal | 3-15 | 4 | 8 |
| 326_50 | MBC | 3-30 | 5-12 | 4 | Non-clonal | 3-15 | 5 | 9 |
| 327_32 | MBC | 3-23 | 3-10 | 4 | Non-clonal | 3-15 | 4 | 8 |
| 334_52 | MBC | 3-48 | 6-13 | 6 | 5 | 3-15 | 4 | 8 |
| 334_53 | MBC | 3-23 | 2-2 | 4 | Non-clonal | 3-15 | 5 | 9 |
| 334_62 | MBC | 3-23 | 2-2 | 1 | Non-clonal | 3-11 | 4 | 6 |
| 337_32 | MBC | 3-30 | 5-24 | 4 | Non-clonal | 3-11 | 5 | 7 |
| 337_43 | MBC | 3-23 | 5-24 | 4 | Non-clonal | 3-15 | 4 | 8 |
| 337_95 | MBC | 3-48 | 3-16 | 6 | Non-clonal | 3-15 | 5 | 9 |
| 346_54 | MBC | 3-30-3 | 3-16 | 5 | Non-clonal | 3-11 | 4 | 6 |
| 347_64 | MBC | 3-23 | 1-7 | 2 | Non-clonal | 3-11 | 5 | 7 |
| 347_246 | MBC | 3-23 | 1-1 | 5 | Non-clonal | 3-15 | 5 | 9 |
| 350_8 | MBC | 3-23 | 3-16 | 4 | Non-clonal | 3-15 | 4 | 8 |
| 350_22 | MBC | 3-23 | 6-19 | 4 | Non-clonal | 3-11 | 4 | 6 |
| 350_132 | MBC | 3-48 | 1-26 | 4 | Non-clonal | 3-15 | 4 | 8 |
| 350_145 | MBC | 3-48 | 4-17 | 6 | 5 | 3-15 | 4 | 8 |
| 351_38 | MBC | 3-23 | 6-13 | 4 | Non-clonal | 3-15 | 4 | 8 |
| 351_93 | MBC | 3-23 | 1-1 | 4 | Non-clonal | 3-15 | 4 | 8 |
| 351_139 | MBC | 3-23 | 5-24 | 4 | Non-clonal | 3-11 | 4 | 6 |

**Extended Data Table 3 | Mutation information for Extended Data Fig. 2f, g**

| Mutation | Naturally Occurring? | 045-09 2B06 Deep Mutational Scan? | Other Experimentally Determined? | Note | Source |
|---|---|---|---|---|---|
| H38S (HA1) | | No | Yes | Affects C179 binding | [42] |
| L292R (HA1) | | Yes | | In footprint of many VH1-69 utilizing mAbs | [43] |
| I323V (HA1) | Yes | No | | Present in pH1N1 viruses | [44] |
| K368D (HA2) | | Yes | Yes | Contacts with VH3-30 mAb | [45] |
| Q371E (HA2) | | Yes | Yes | Interact with CR9114 and FI6v3 | [46] |
| A373V (HA2) | | No | Yes | Expands in presence of stalk binding mAbs, escape mutant of 6F12 | [2,47] |
| I374L (HA2) | | Yes | Yes | Mutation associated with resistance to group 1 neutralizing stalk mAbs | [46,48,49] |
| D375R (HA2) | | Yes | | Pre-pH1N1 viruses express N375 | |
| E376K (HA2) | Yes | Yes | | Associated with HA protein stability | [50] |
| T378R (HA2) | | Yes | | Mutation associated with resistance to group 1 neutralizing stalk mAbs | [48,49] |
| N379R (HA2) | | Yes | | Interacts with CR9114 | [10] |
| V381A (HA2) | | No | Yes | Affects FI6v3 mAb binding | [46] |
| I385A (HA2) | | No | Yes | Affects F10 mAb binding | [51] |
| E386K (HA2) | | Yes | | Critical residue for membrane fusion | [52] |

**Extended Data Table 4 | Cryo-EM data collection and refinement statistics for 222-1CO6 binding H1. Related to Fig. 2**

|  | (EMDB-25655)<br>(PDB 7T3D) |
|---|---|
| **Data collection and processing** | |
| Magnification | 36,000 |
| Voltage (kV) | 200 |
| Electron exposure (e–/Å$^2$) | 49.9 |
| Defocus range (µm) | -0.8 to -1.8 |
| Pixel size (Å) | 1.15 |
| Symmetry imposed | C3 |
| Initial particle images (no.) | 92,514 (post 2D classification) |
| Final particle images (no.) | 48,846 |
| Map resolution (Å) | 3.38 |
|   FSC threshold | 0.143 |
| Map resolution range (Å) | 3.2-4.4 |
| | |
| **Refinement** | |
| Initial model used (PDB code) | 7MEM |
| Model resolution (Å) | 3.5 |
|   FSC threshold | 0.5 |
| Map sharpening $B$ factor (Å$^2$) | -88.1 |
| Model composition | |
|   Non-hydrogen atoms | 22122 |
|   Protein residues | 2796 |
|   Ligands | NAG:21 |
| R.m.s. deviations | |
|   Bond lengths (Å) | 0.022 |
|   Bond angles (°) | 1.732 |
| Validation | |
|   MolProbity score | 0.9 |
|   Clashscore | 1.54 |
|   Poor rotamers (%) | 0 |
| Ramachandran plot | |
|   Favored (%) | 98.40 |
|   Allowed (%) | 1.96 |
|   Disallowed (%) | 0.00 |

# Reporting Summary

## Statistics

For all statistical analyses, confirm that the following items are present in the figure legend, table legend, main text, or Methods section.

| n/a | Confirmed | |
|---|---|---|
| ☐ | ☒ | The exact sample size (*n*) for each experimental group/condition, given as a discrete number and unit of measurement |
| ☐ | ☒ | A statement on whether measurements were taken from distinct samples or whether the same sample was measured repeatedly |
| ☐ | ☒ | The statistical test(s) used AND whether they are one- or two-sided<br>*Only common tests should be described solely by name; describe more complex techniques in the Methods section.* |
| ☒ | ☐ | A description of all covariates tested |
| ☒ | ☐ | A description of any assumptions or corrections, such as tests of normality and adjustment for multiple comparisons |
| ☐ | ☒ | A full description of the statistical parameters including central tendency (e.g. means) or other basic estimates (e.g. regression coefficient) AND variation (e.g. standard deviation) or associated estimates of uncertainty (e.g. confidence intervals) |
| ☐ | ☒ | For null hypothesis testing, the test statistic (e.g. *F*, *t*, *r*) with confidence intervals, effect sizes, degrees of freedom and *P* value noted<br>*Give P values as exact values whenever suitable.* |
| ☒ | ☐ | For Bayesian analysis, information on the choice of priors and Markov chain Monte Carlo settings |
| ☒ | ☐ | For hierarchical and complex designs, identification of the appropriate level for tests and full reporting of outcomes |
| ☒ | ☐ | Estimates of effect sizes (e.g. Cohen's *d*, Pearson's *r*), indicating how they were calculated |

*Our web collection on statistics for biologists contains articles on many of the points above.*

## Software and code

Policy information about availability of computer code

| Data collection | Software - Leginon beta and v3.2, Appion beta |
|---|---|
| Data analysis | Code - dms_tools2 version 2.4.12, dms_tools version 1.1.12<br>Software - Prism 8, Prism 9, VGene (beta), Librator (beta), Cellrander Single-Cell Software suite v3.0, Seurat 3 v3.2.0, LinQ-View v0.99, WebLogo v2.8.2, UCSF Chimera 1.14, Pymol v2.3.4, Rosettaantibody v2021, MOE v2020, AMBER v20, GROMACS 2019.2 with plumed-2.5.2., PyEMMA v2.5.7, FlowJo v10, MotionCor v2, Relion v3.0, XQuartz v2.7.11, abYsis v3.4.1, Rosetta 2020.03.61102, coot 0.9-pre EL, CryoSPARC2 3.2.0, Phenix 1.17.1-3660, Clustal Omega (https://www.ebi.ac.uk/Tools/msa/clustalo). |

For manuscripts utilizing custom algorithms or software that are central to the research but not yet described in published literature, software must be made available to editors and reviewers. We strongly encourage code deposition in a community repository (e.g. GitHub). See the Nature Portfolio guidelines for submitting code & software for further information.

## Data

Policy information about availability of data

All manuscripts must include a data availability statement. This statement should provide the following information, where applicable:

- Accession codes, unique identifiers, or web links for publicly available datasets
- A description of any restrictions on data availability
- For clinical datasets or third party data, please ensure that the statement adheres to our policy

Repertoire data generated from single cell RNA-sequencing data is deposited at Mendeley Data (https://data.mendeley.com/datasets/jzsx489pmk/1). Accession numbers for all other anchor-targeting mAbs are included in Supplemental Table 1. Electron microscopy maps were deposited to the Electron Microscopy DataBank under accession IDs: D_100025433, D_1000254374, D_1000254375, D_1000254376, D_1000254377, D_1000254378, D_1000254383, D_1000254384,

D_1000254385, D_1000254386, D_1000254388, D_1000254379, D_1000254391, and D_1000254382. All next generation sequencing data for 045-09 2B06 deep mutational scanning and for the H1N1 mutational scanning can be found on the Sequence Read Archive under BioProject accession number PRJNA309339. The following Protein Database accession numbers were downloaded and included in the manuscript - 6HJQ, 3SDY, 4NM8, 6HJQ, 4M4Y, 4WE4, 4JTV, 4FQI, 3ZTN, and 7MEM.

# Field-specific reporting

Please select the one below that is the best fit for your research. If you are not sure, read the appropriate sections before making your selection.

☒ Life sciences ☐ Behavioural & social sciences ☐ Ecological, evolutionary & environmental sciences

For a reference copy of the document with all sections, see nature.com/documents/nr-reporting-summary-flat.pdf

# Life sciences study design

All studies must disclose on these points even when the disclosure is negative.

| | |
|---|---|
| Sample size | Samples sizes were based on the number of donors and ability to process samples. |
| Data exclusions | No data were excluded. |
| Replication | All experiments were performed more than once, except the mouse viral titers. All replications were successful. |
| Randomization | Recipients for the cHA vaccine arms were randomized. Otherwise, no other experimental groups were used and therefore were not randomized. |
| Blinding | Serum competition ELISAs in Fig. 4 and Extended Data 6 were blinded. Otherwise, no other blinding was performed. |

# Reporting for specific materials, systems and methods

We require information from authors about some types of materials, experimental systems and methods used in many studies. Here, indicate whether each material, system or method listed is relevant to your study. If you are not sure if a list item applies to your research, read the appropriate section before selecting a response.

## Materials & experimental systems

| n/a | Involved in the study |
|---|---|
| ☐ | ☒ Antibodies |
| ☐ | ☒ Eukaryotic cell lines |
| ☒ | ☐ Palaeontology and archaeology |
| ☐ | ☒ Animals and other organisms |
| ☐ | ☒ Human research participants |
| ☐ | ☒ Clinical data |
| ☒ | ☐ Dual use research of concern |

## Methods

| n/a | Involved in the study |
|---|---|
| ☒ | ☐ ChIP-seq |
| ☐ | ☒ Flow cytometry |
| ☒ | ☐ MRI-based neuroimaging |

# Antibodies

| | |
|---|---|
| Antibodies used | Anti-human IgG Fc - BV421, clone M1310G05, Biolegend, Cat#410703<br>Mouse anti-influenza A nucleoprotein antibody, clone A3, biotin, Sigma/Millipore, MAB8258B-5<br>Polyclonal Goat Anti-human IgG HRP, Jackson Immunoresearch, Cat#109-035-098<br>Polyclonal Goat anti-mouse IgG HRP, Southern Biotech, Cat# 1030-05<br>Polyclonal Goat anti-mouse IgG H&L peroxidase-conjugated, Rockland Cat#610-1302<br><br>Monoclonal antibodies from previous publications and synthesized in-house are listed below with reference -<br>CR9114 - 10.1126/science.1222908<br>3H9 - 10.1073/pnas.84.24.9150<br><br>Monoclonal antibodies unique to this study and synthesized in-house are listed below -<br>030-09 3E05<br>030-09M 1B06<br>045-09 1A03<br>045-09 2B03<br>047-09 4F04<br>SFV009 3D04<br>SFV009 3G01<br>SFV009 3G03<br>236 IgG 1A02 |

236 IgG 1D01
236 IgG 1F01
236 IgA 1F06
241 IgG 2A06
241 IgA 1D05
241 IgA 2F04
241 IgA 2F06
121 2C06
222 1C06
301_91
301_249
301_275
308_60
310_10
310_49
317_30
317_117
319_75
319_147
319_345
319_373
322_48
324_5
326_38
326_42
326_50
327_32
334_52
334_53
334_62
337_32
337_43
337_95
346_54
347_64
347_246
350_8
350_22
350_132
350_145
351_38
351_93
351_139
029-09 4A01
047-09 1F05
047-09 4E01
051-094B02
051-09 4C06
051-09 4E06
051-09 5A02
051-09 5C01
030-09 1A06
030-09 1E04
030-09 2B03
030-09 2G03
030-09 3B03
045-09 2B06
SFV005 2G02
SFV019 4E03
SFV009 2A06
SFV009 3A01
sc1009 3B05
sc1009 3E06
SF1000 3D04
sc70 1F02
sc70 5B03
039-10 5F06
039-10 5G02
039-10 5G05
240 IgG 1C04
220 IgG 1A05
237 IgG 1D01
241 IgA 1D05
241 IgA 2E06
301-48
301-90

301-277
308-2
311-30
319-4
319-73
319-99
319-256
319-418
324-184
334-98
337-51
337-53
347-58
347-140
351-31

Validation | All commercial antibodies were validated by their manufacturers and were titrated in the lab to determine optimal concentration for experimentation. In-house generated monoclonal antibodies were validated in preliminary ELISAs to A/California/7/2009 H1N1 virus or recombinant H1 or cH5/1 proteins. MAb concentrations were standardized based on the assay and starting concentration is described in methods section.

## Eukaryotic cell lines

Policy information about cell lines

Cell line source(s) | HEK293T cells - ATCC
MDCK cells - ATCC
A549 cells - ATCC
Jurkat cells expressing FcgRIIIa with NFAT-drive luciferase reporter gene - Promega, G7010
MDCK-SIAT1 - generated in 10.1126/science.1187816.

Authentication | Cell lines were authenticated by supplier. No other authentication at the lab level was performed.

Mycoplasma contamination | Cell lines were not tested.

Commonly misidentified lines (See ICLAC register) | No commonly misidentified cell lines were used in this study.

## Animals and other organisms

Policy information about studies involving animals; ARRIVE guidelines recommended for reporting animal research

Laboratory animals | 6-8 week old female Balb/c mice were used for challenge studies. Mice were housed in ABSL-2 conditions with 12-hour light/dark cycles and controlled temperature and humidity.

Wild animals | No wild animals were used in this study

Field-collected samples | No field collected samples were used in this study.

Ethics oversight | Animal experiments were approved by the University of Chicago and Icahn School of Medicine at Mount Sinai IACUCs.

Note that full information on the approval of the study protocol must also be provided in the manuscript.

## Human research participants

Policy information about studies involving human research participants

Population characteristics | For the 2009 MIV, 2009 pH1N1 infection, 2010 TIV, and 2014 QIV cohorts, median age was 30 years old with a range of 20-64. 63% of donors were women. Only birth date, prior influenza virus vaccination status, and sex were obtained or released to the authors for donors within these cohorts.

For the cHA vaccine trial - In Groups 1, 2, and 4 about two-thirds of the subjects were female (70.0%, 66.7%, and 62.5%, respectively), compared to 2/5 (40.0%) in Group 3 and 5/10 (50.0%) in Group 5. Most were non-Hispanic or Latino (80% to 100% per group) and black or African American (66.7% to 87.5%). The median age at enrollment ranged from 26 to 31 years across groups, with minimum age ranging from 18 to 22 and maximum age from 29 to 38. The median weight ranged from 74.7 kg to 84.2 kg and the median height ranged from 165.5 cm to 174.0 cm.
The most common pre-existing conditions across all treatment groups were immune system disorders (24/66, 36.4%), the majority being allergies (seasonal and food), followed by drug hypersensitivity.
No subjects in the placebo Groups 3 and 5 reported taking prior medications and more subjects in Group 1 (5/19, 26.3%) reported taking prior medications compared to Group 2 (1/14, 7.1%) and Group 4 (2/15, 13.3%). The proportions of subjects taking concomitant medications in Groups 1 to 5 were 84.2%, 78.6%, 100%, 60.0% and 80.0%, respectively. The most common, by therapeutic subgroup, were analgesics (20/61, 32.8%), anti-inflammatory and anti rheumatic products (19/61, 31.1%) and sex hormones and modulators of the genital system (18/61, 29.5%).

| Recruitment | Participants were recruited from the local community for all cohorts. The target population reflected the community at large. All participants provided informed consent. Donors for the 2009 MIV, 2010 TIV, 2014 QIV, and 2009 pH1N1 infection had been recruited from prior studies from the local community. Participants for the cHA vaccine trial were recruited according to Bernstein et al. Lancet Infectious Disease 2020 and Nachbagauer et al. Nature Medicine 2020.The target population reflected the community at large at each of the participating study sites. Information regarding this trial was provided to potential subjects who have previously participated in vaccine trials conducted at the participating study sites. |
|---|---|
| Ethics oversight | The study protocol was approved by the IRB at the University of Chicago for all studies, the Emory University for the 2009 pH1N1 infection and 2009 MIV cohorts, and at Icahn School of Medicine at Mount Sinai, Duke University and Cincinnati Children's Hospital Medical Center (CCHMC) for the cHA vaccine study. |

Note that full information on the approval of the study protocol must also be provided in the manuscript.

# Clinical data

Policy information about clinical studies

All manuscripts should comply with the ICMJE guidelines for publication of clinical research and a completed CONSORT checklist must be included with all submissions.

| Clinical trial registration | NCT03300050 |
|---|---|
| Study protocol | Study Protocol can be accessed here - https://clinicaltrials.gov/ProvidedDocs/50/NCT03300050/Prot_000.pdf |
| Data collection | Gamble Program for Clinical Studies, Cincinnati Children's Hospital Medical Center 3333 Burnet Avenue, Cincinnati, OH 45229-3039<br>Duke Early Phase Clinical Research Unit, Duke Clinical Research Institute 40 Duke Medicine Circle, Durham, NC 27710<br>Studied period:<br>Date of first enrollment: 10 October 2017<br>Date of last subject completion: 09 August 2019 |
| Outcomes | Primary Objectives<br>To assess the reactogenicity and safety through 28 days after each priming dose of cH8/1Nl LAIV (or placebo) and the booster dose of cH5/1Nl IIV +/- AS03A (or placebo) and through 28 days after each dose of IIV (cH8/1Nl IIV + AS03A and cH5/1Nl IIV + AS03A) (or placebo) in terms of rates of solicited local and general adverse events (AEs) through 7 days post-vaccination, unsolicited AEs through 28 days post vaccination, hematological and biochemical laboratory abnormalities up to Visit 13, and medically attended event (MAEs), laboratory-confirmed influenza-like illness (LC-ILi), potential immune-mediated disease (pIMDs), and serious adverse events (SAEs) through Visit 13.<br>Secondary Objectives<br>For this study, serum antibody titers (EC50) for the anchor and central stalk epitopes was determined. Only individuals with samples at d1, 28, 85, 113, and 420 were included in the analysis. Seroconversion was considered when titers increased by 1.5x over pre-vaccine time points (d1 or d85). |

# Flow Cytometry

## Plots

Confirm that:

☒ The axis labels state the marker and fluorochrome used (e.g. CD4-FITC).

☒ The axis scales are clearly visible. Include numbers along axes only for bottom left plot of group (a 'group' is an analysis of identical markers).

☒ All plots are contour plots with outliers or pseudocolor plots.

☒ A numerical value for number of cells or percentage (with statistics) is provided.

## Methodology

| Sample preparation | HEK293T cells were transfected with plasmids to express full length membrane-bound A/California/7/2009 H1 or membrane bound mini-HA. Cells were harvested 4 days later and stained with monoclonal antibodies of interest, which were detected with an anti-human IgG-BV421. |
|---|---|
| Instrument | BD LSRFortessa |
| Software | Data Collection - FACSDIva<br>Anaylsis - FlowJo v10 |
| Cell population abundance | Cell population abundance is shown for representative panels and gates are identical across individual datasets. |
| Gating strategy | Cells were gated on using FSC and SSC. From this gate, mAb+ cells were gated on, based on the secondary Ab only stain. Gating strategy and examples are in Supplemental Fig. 1. For mAb+ gating, histograms are represented. |

☒ Tick this box to confirm that a figure exemplifying the gating strategy is provided in the Supplementary Information.

