## [Peer Review File · Nature]

Manuscript Title: Broadly neutralizing antibodies target a hemagglutinin anchor epitope

Reviewer Comments & Author Rebuttals

Reviewer Reports on the Initial Version:

Referee #1

In the manuscript, the authors identify a class of antibodies that recognize the membrane-proximal region of the viral hemagglutinin (HA) and comprehensively characterize them through structural, immunogenetic, and functional analyses. Although the overwhelming majority of the anchor-targeting antibodies are H1 subtype-specific, several antibodies show cross-reactivity beyond H1 to other group 1 HA subtypes. Interestingly, the anchor-targeting antibodies appear to possess much weaker ADCC activity compared to the central stalk-targeting antibodies, while both anchor- and central stalk-targeting antibodies provide comparable protection to lethal H1N1 infection in mice in a passive prophylaxis setting. The authors demonstrated that the anchor-targeting antibodies were readily detectable in individuals upon immunization with commercial vaccines by EMPEM or with the experimental vaccine containing adjuvanted chimeric HA by serum antibody competition. As the authors rightly mentioned, the first antibody that targets the anchor epitope (FISW84) was previously reported (Benton et al. PNAS, 2018). Indeed, not surprisingly, the FISW84 is H1-specific and possesses the canonical immunogenetic signatures—VH3-23+VK3-15 with NWP motif—described in this manuscript. The present study, however, identified the existence of the anchor-targeting antibodies in a fairly large fraction in the H1-specific population across multiple individuals, and those antibodies are genetically constrained with unique immunogenetic signatures. This class of antibodies appears to be vaccine-inducible and contributing to a serum antibody response to HA. While the present study elegantly showcases the high points of this class of antibodies, implications that would lead to a new vaccine and/or therapeutic concepts, as well as broader impact to advance the field, are somewhat limited due in part to the lack of novelty of the anchor epitope and its relatively narrow specificity. My specific comments follow.

Major comments:

1. How are the neutralization curves of anchor-targeting mAbs compared to that of central stalk-targeting mAbs? Although it does not reach statistical difference, the neutralization potency of anchor-targeting mAbs seems slightly lower than central stalk-targeting mAbs (Fig. 1e). Are the IC80 values of anchor- and central stalk-targeting mAbs also comparable?
2. Since the manuscript title suggests that the anchor-targeting mAbs are broadly neutralizing, the authors should show heterosubtypic H2N2 and potentially other group 1 virus neutralization data (line 84, data not shown). Otherwise, this reviewer suggests the authors use the term “pan-H1” or “broad H1” instead to appropriately reflect the findings.
3. MD simulation of the NWP motif of 3 anchor mAbs (ED Fig. 3m) rather shows the diversity of this motif in interacting with anchor epitope residues (or adapting to different CDR L3 loop conformations). Orientations of N93, W94, and P95 look somewhat random. It would be interesting and more informative to compare structures of 222-1C06 with FISW84 (PDB 6HKG) in my opinion.
4. Are the comparisons in Fig. 4d and 4e between adjuvanted chimeric HA-containing IIV and unadjuvanted commercial IIV? If so, it is very misleading. The proper comparator should be adjuvanted commercial IIV (or unadjuvanted cHA-containing IIV). It is more likely that the observed differences could just be attributed to the use of an adjuvant.
5. A few of group 2 HA-specific mAbs such as CR8020 and CR8043 also target the membrane-proximal epitope. Comparison of the anchor epitope with the theoretical footprint of these group 2 mAbs may provide insights for the anchor epitopes in general. Although the group 2 mAbs have a quite different angle of approach, it is of interest to compare and contrast them with ones discussed in the manuscript. The discussion on the unusual 'upward' angle of approach of the anchor-targeting mAbs reported here would be helpful.

6. Are the anchor-targeting mAbs poly- or autoreactive?
7. It is quite intriguing to see the sharp contrast between anchor- and central stalk-targeting mAbs on ADCC activity (ED Fig. 2e and 2f). Can this be explained by the positioning of Fc and distance to the effector cells that some of the authors on this manuscript previously proposed (head vs. stalk)? It would be informative to have potential explanations or speculation on this point.
8. Since the anchor-targeting mAbs can recognize both pre-pandemic and pandemic H1 lineages it would be very interesting to retrospectively analyze the anchor-targeting B cell response when pH1N1 was introduced. The H-L Ig paired sequencing datasets in DeKosky et al. (Nat Biotechnol, 2013) may have some clues to address this question.

Minor comments:

1. It is not clear to me how the authors determine the HA domain specificities (line 68 & ED Fig. 1a). All 358 mAbs are isolated by the same H1 HA probe?
2. The term "multiple donors" in lines 109-110 is misleading and should be replaced with "two donors".
3. Why did the authors give mAbs only 2 hours before the virus challenge in mice? There will not be enough time for IP-injected mAbs to be distributed in the airway and lung to prevent virus infection.
4. Did the authors assess the escapability of the anchor epitope? It is probably possible to infer the fitness of the virus possessing mutations within the anchor epitope from existing DMS data.

Referee #2

A. Summary of key results

This study by Guthmiller et al. explored a distinct class of broadly-neutralizing antibodies that target the influenza hemagglutinin (HA) at the anchor epitope, a site at the base of the HA stalk near its membrane anchor. A panel of 358 human monoclonal antibodies (mAbs) was generated, and 50 mAbs deriving from 21 individuals target the anchor epitope. The anchor-epitope-targeting mAbs bind and neutralize diverse H1 influenza viruses from the past 40 years, as well as several mutant HAs with known central-stalk-epitope escape mutants. Electron microscopy structural studies confirm that the epitope is distinct from the central-stalk epitope, and mAbs make extensive contacts across the HA fusion peptide. Interestingly, the antibodies all bind in an "upward" angle, suggesting that the epitope may only be transiently exposed during HA flexing on the membrane, and possibly explaining the absence of ADCC observed by the anchor-epitope-targeting mAbs due to this angle of HA binding. Importantly, prophylactic studies in mice showed 100% protection by anchor-epitope-targeting mAbs from a lethal dose of H1N1 virus, similar to the protection seen by central stalk mAbs. Notably, sequence analyses of the anchor-epitope-targeting mAbs reveal highly restricted heavy and light chain repertoires, with one public clone identified across multiple participants. Moreover, of the 22 participants who received chimeric HA vaccines, which boosts HA stalk-directed antibodies, the large majority of participants generated B cells with anchor-binding antibody restricted repertoires. Electron microscopy polyclonal epitope mapping and competition ELISAs confirmed the presence and abundance of anchor-epitope-targeting antibodies in serum samples from these chimeric HA vaccine participants. Overall, this study uncovered the existence and protective potential of a public class of broadly neutralizing antibodies that target the conserved HA anchor epitope.

B. Originality and significance

This study by Guthmiller et al. substantially enhances our understanding and provides new context to two previous publications:

1. "Influenza hemagglutinin membrane anchor" by Benton et al. in PNAS in 2018.
DOI:10.1073/pnas.181092711

In this publication, the human monoclonal antibody FISW84 was shown via cryoEM studies to bind at an "upward" angle to a conserved epitope at the base of the HA stalk near the "membrane anchor." FISW84 broadly neutralizes diverse H1 influenza viruses. While this study showed the first existence of the HA anchor epitope, it was unclear how rare anchor-epitope-

targeting antibodies are in humans and whether this class of antibodies is protective in vivo. The current study by Guthmiller et al. revealed that these antibodies are common in humans and are protective in vivo.

2. "A chimeric hemagglutinin-based universal influenza virus vaccine approach induces broad and long-lasting immunity in a randomized, placebo-controlled phase I trial." By Nachbagauer et al. in Nat. Med. in 2021.

DOI:10.1038/s41591-020-1118-7

In this publication, volunteers received vaccinations with sequential chimeric HA vaccines, which redirects immune responses to the HA stalk region. Anti-stalk antibody levels were boosted to high titers. Passive transfer of vaccinated human serum antibodies into mice showed protection from weight loss induced by an H6 virus that would only be reactive to anti-stalk antibodies. While this study showed that chimeric HA vaccines could boost HA stalk-directed antibody levels, it was unclear whether these antibodies targeted the central stalk epitope or the anchor epitope. The current study by Guthmiller et al. revealed that chimeric HA vaccines do indeed induce antibodies against the anchor epitope in the majority of vaccinated individuals, and that anchor-epitope-targeting antibodies specifically are protective.

Thus, while the current study is not the first to describe the HA anchor epitope, it is the first to show that most people can develop antibodies against this epitope, the first to show that antibodies against this epitope are protective in vivo, and the first to show that chimeric HA vaccines induce antibodies against this epitope. Therefore, the current study provides the necessary knowledge and context to target this HA epitope in the development of universal flu vaccines.

C. Data and methodology

This study was a massive undertaking in experimentation. The data was robust and of high quality. There are some minor suggestions for improvements to data presentation (described below).

D. Appropriate use of statistics

Statistics appear appropriate.

E. Conclusions

The conclusions are well justified by the data presented.

F. Suggested improvements

- Lines 65 – 67 and Ext. Fig. 1a: This section is a bit misleading, as it states "vaccinated and naturally infected volunteers" but does not clarify that ~1/3 of volunteers were from a chimeric HA vaccine cohort, whose immune responses are intentionally directed by vaccination toward the HA stalk. Thus, saying that "nearly half" of the mAbs were directed at the HA stalk is misleading, as this would not be expected from a normal group of people who were naturally infected or received a seasonal flu vaccine. (Indeed, of the 52 anchor-epitope-targeting mAbs isolated in this study, 44 of them were derived from chimeric HA vaccine volunteers). It is recommended that the language be clarified, and to consider deleting the left half of Extended figure 1a, which is specific to this unique cohort.

- Line 152: "the anchor epitope is a common target of the human MBC repertoire against HA" should be clarified as "the anchor epitope is a common target of the human MBC repertoire following chimeric HA vaccination."

- Line 160: "the anchor epitope is common within the MBC pool and polyclonal serum antibody

response” should be clarified as “the anchor epitope is common within the MBC pool and polyclonal serum antibody response in chimeric HA vaccinated individuals.”

- Lines 162-174, Figure 4b,c, and Ext. Fig. 5c,d: the jargon should be simplified, as many people may not know what IIV, MIV, AS03, cH5/1, or cH8/1 mean, and it isn't really necessary to explain this. Instead, it is recommended to simplify to “adjuvanted cHA” to describe the results. Same with labels in Figure 4b,c, and Ext. Fig. 5c,d and their legends.

- Figure 2e: lavender color is hard to see – it is suggested to change to another color

- Ext. Table 1: should include footnotes to explain what the different vaccines are.

- Line 463: “in process of being deposited” – data should be deposited and made available by publication.

G. References

- References 5 and 13 appear to be the same reference.

H. Clarity and context

Abstract, introduction, and conclusions are appropriate.

Referee #3

This manuscript focuses on novel HA anchor epitope-binding antibodies. The authors first generated 358 mAbs from vaccinated and naturally infected volunteers and classified them according to their recognized epitopes. They identified 50 mAbs that bind to the anchor epitope of the HA stalk. They revealed that all of the identified anchor epitope-binding mAbs utilized unique repertoires that differ from those of Central Stalk epitope-binding mAbs. In addition, the authors showed that anchor epitope-binding B cells are common among human memory B cells and a chimeric HA vaccine strategy recalls/activates these cells, suggesting that induction of these antibodies is key for protection against seasonal and pandemic influenza viruses. The manuscript is well written and the data are plentiful and well presented. However, the authors need to make it clearer that anchor epitope-binding antibodies are superior to previously identified broadly neutralizing antibodies. Also, some issues need to be addressed, as detailed below.

Major points:

1. Overall, the anchor epitope-binding antibodies are novel, but the binding affinities of these antibodies are less broad than those of Central Stalk epitope-binding mAbs.

The authors should show and discuss the superiority of anchor epitope-binding antibodies compared with previously identified Central Stalk epitope-binding mAbs.

2. Page 5, lines 98-100. Three anchor epitope-binding mAbs possess ADCC activity, as well as Central Stalk epitope-binding mAbs. What causes the different ADCC activity among the anchor epitope-binding mAbs?

3. Page 9, line 198. The authors suggested that anti-anchor antibodies have the potential to cross-neutralize group 1 viruses including H2 and H5. However, as shown in Fig. 1f, anti-anchor antibodies that can bind to H2 or H5 are quite rare. Although it is clear that a few of the anti-anchor antibodies cross-neutralize H2 or H5, the anchor epitope does not seem to be an important target for bnAbs against group 1 viruses. Therefore, the authors' discussion on this point is a little misleading.

4. Fig. 4. The authors should examine neutralizing titers against various H1 viruses. When serum neutralizes various H1 viruses, it is important to show how large the impact of anchor epitope-

binding Abs is for the neutralization compared with Central Stalk epitope-binding Abs.

5. Extended Data. Fig. 2. The authors should examine the protective effect in terms of the therapeutic use of these anchor-epitope binding antibodies. Also, to investigate the potential of anchor epitope-targeting antibodies to provide protection against pan-H1 influenza virus, the authors should use multiple virus strains.

Minor points:

1. Please provide a reference for CR9114 and FI6v3.
2. Fig. 1c. It is hard to distinguish between the colors used for mapping.
3. Fig. 1f. The authors should state how they defined mAbs binding.
4. Line 84. Please show the data, which would be informative.
5. Extended Data Fig. 1h and lines 90-91. The effect of the HA2 A44V mutant on the binding to the HA stalk is larger than that of the other stalk mutants. The authors should comment on this.
6. Extended Data. Fig. 2. To test whether mAbs targeting the anchor epitope were protective in vivo, the authors provided only the data of weight loss and survival of infected mice. However, viral titers in organs should also be tested to demonstrate the protective efficacy of the mAbs.
7. Fig. 4 and Extended Data Fig. 5 a-d. The labels of the X-axis are a little confusing for readers. Please provide sufficient explanation of the abbreviations such as IIV and LAIV.
8. Fig. 4e. please spell out "EC50".
9. Fig. 5d, Y-axis. Characters are overlapping. Please correct.
10. Materials and Methods, page 2, lines 38-42. It is not appropriate for several authors to be acknowledged in this section.

Author Rebuttals to Initial Comments:

Referee #1 (Remarks to the Author):

In the manuscript, the authors identify a class of antibodies that recognize the membrane-proximal region of the viral hemagglutinin (HA) and comprehensively characterize them through structural, immunogenetic, and functional analyses. Although the overwhelming majority of the anchor-targeting antibodies are H1 subtype-specific, several antibodies show cross-reactivity beyond H1 to other group 1 HA subtypes. Interestingly, the anchor-targeting antibodies appear to possess much weaker ADCC activity compared to the central stalk-targeting antibodies, while both anchor- and central stalk-targeting antibodies provide comparable protection to lethal H1N1 infection in mice in a passive prophylaxis setting. The authors demonstrated that the anchor-targeting antibodies were readily detectable in individuals upon immunization with commercial vaccines by EMPEM or with the experimental vaccine containing adjuvanted chimeric HA by serum antibody competition. As the authors rightly mentioned, the first antibody that targets the anchor epitope (FISW84) was previously reported (Benton et al. PNAS, 2018). Indeed, not surprisingly, the FISW84 is H1-specific and possesses the canonical immunogenetic signatures—VH3-23+VK3-15 with NWP motif—described in this manuscript. The present study, however, identified the existence of the anchor-targeting antibodies in a fairly large fraction in the H1-specific population across multiple individuals, and those antibodies are

genetically constrained with unique immunogenetic signatures. This class of antibodies appears to be vaccine-inducible and contributing to a serum antibody response to HA. While the present study elegantly showcases the high points of this class of antibodies, implications that would lead to a new vaccine and/or therapeutic concepts, as well as broader impact to advance the field, are somewhat limited due in part to the lack of novelty of the anchor epitope and its relatively narrow specificity. My specific comments follow.

Major comments:

1. How are the neutralization curves of anchor-targeting mAbs compared to that of central stalk-targeting mAbs? Although it does not reach statistical difference, the neutralization potency of anchor-targeting mAbs seems slightly lower than central stalk-targeting mAbs (Fig. 1e). Are the IC80 values of anchor- and central stalk-targeting mAbs also comparable?

This is an interesting and important point raised by the reviewer. Generally, we find a wide variety of IC50s and IC80s (nearly 100-fold differences from least to most potent) for antibodies targeting both the anchor and central stalk epitopes. We have included the neutralization curves and IC80s for anchor and central stalk-binding mAbs to Extended data Fig. 1g-h. Similar to the IC50 data, there were differences in IC80 potency and there was actually a shift towards anchor-binding mAbs being slightly more potent, albeit not statistically significant.

2. Since the manuscript title suggests that the anchor-targeting mAbs are broadly neutralizing, the authors should show heterosubtypic H2N2 and potentially other group 1 virus neutralization data (line 84, data not shown). Otherwise, this reviewer suggests the authors use the term “pan-H1” or “broad H1” instead to appropriately reflect the findings.

We have added H2 neutralization data to Extended Data Fig. 2c, where 2 of the 11 tested anchor-binding mAbs neutralized H2. Of note, the term broadly neutralizing has been used to refer to antibodies that can neutralize both pre- and post-pH1N1 viruses, therefore, we believe this terminology is still accurate. However, we have added in at line 132 that these antibodies are pan-H1 neutralizing.

3. MD simulation of the NWP motif of 3 anchor mAbs (ED Fig. 3m) rather shows the diversity of this motif in interacting with anchor epitope residues (or adapting to different CDR L3 loop conformations). Orientations of N93, W94, and P95 look somewhat random. It would be interesting and more informative to compare structures of 222-1C06 with FISW84 (PDB 6HKG) in my opinion.

We agree that comparing the two structures is of greater interest than the MD simulations and have added these data to Extended data fig. 3m, which show similar orientations of the aromatic pocket of the K-CDR3 and Y59 of the H-CDR2. The overall resolution of FISW84 was $> 4 \text{ \AA}$, which precluded resolution of

the precise molecular contacts. Despite this, it is clear that the NWP of the K-CDR3 and the Y59 of the H-CDR2 of FISW84 are facing towards the fusion peptide.

In addition, we have clarified that the MD simulations were meant to show the aromatic pockets of the VH3-23 and VH3-30 utilizing mAbs were similarly oriented towards the fusion peptide and therefore likely recognize this epitope through a similar mechanism. We have also added an MD simulation of FISW84 to show that the orientation of this antibody, which utilizes VH3-23/VK3-15, is oriented slightly differently from another mAb in our study that used the same heavy/light chain pairing.

4. Are the comparisons in Fig. 4d and 4e between adjuvanted chimeric HA-containing IIV and unadjuvanted commercial IIV? If so, it is very misleading. The proper comparator should be adjuvanted commercial IIV (or unadjuvanted cHA-containing IIV). It is more likely that the observed differences could just be attributed to the use of an adjuvant.

We completely agree with the reviewer's comment and the point of Fig. 4 d-e was to show how important a role the adjuvant is in inducing antibody responses against the HA stalk domain. We also agree that our data shows the importance of the inclusion of the AS03 adjuvant but not the independent role of the cHA immunogen. Notably, the original study (Bernstein et al. Lancet Infectious Disease 2020) from this trial showed limited plasmablast responses in those participants that received the inactivated vaccine without adjuvant. [10.1016/S1473-3099\(19\)30393-7](https://doi.org/10.1016/S1473-3099(19)30393-7)

To correct this interpretation, we have added these points starting at line 214 in the results and included in the discussion at line 271 that further research is needed on whether the addition of oil-in-water adjuvants to seasonal vaccines could similarly induce antibodies against distinct epitopes of the stalk domain. In addition, in the abstract, we have added that it was the adjuvanted cHA vaccine that was capable of inducing antibodies against the anchor epitope.

5. A few of group 2 HA-specific mAbs such as CR8020 and CR8043 also target the membrane-proximal epitope. Comparison of the anchor epitope with the theoretical footprint of these group 2 mAbs may provide insights for the anchor epitopes in general. Although the group 2 mAbs have a quite different angle of approach, it is of interest to compare and contrast them with ones discussed in the manuscript. The discussion on the unusual 'upward' angle of approach of the anchor-targeting mAbs reported here would be helpful.

We appreciate the reviewer's comment on this, because it is of general interest whether an equivalent epitope exists on group 2 viruses. However, these epitopes are quite distinct both in location and angle of approach. We have added a panel to Extended Data Fig. 1e to show the relative epitopes and angle of approach of CR8020 and CR8043, which bind the group 2 stalk epitope, relative to that of an anchor-binding mAb.

Notably, the epitope targeted by CR8020/CR8043 is considerably higher and more central on the protomer than that of the anchor epitope and the angle of approach of CR8020/CR8043 is from above, similar to mAbs targeting the central stalk epitope. Related to point 7 below, this angle of approach of antibodies targeting the group 2 stalk epitope as well as the central stalk epitope are likely to engage FcRs, whereas the anchor targeting antibodies may not be able to due to their angle of approach.

6. Are the anchor-targeting mAbs poly- or autoreactive?

We have added these data to Extended data Fig. 2d-e. Notably, about 60% of anchor-targeting mAbs were polyreactive, whereas 90% of mAbs against the central stalk were polyreactive ($P < 0.0001$). In addition, the anchor mAbs that were polyreactive tended to have lower reactivity with LPS relative to mAbs against the central stalk epitope. Therefore, these data suggest that polyreactivity is selected into the anti-anchor B cell repertoire but not to the same degree as those against the central stalk epitope. We have added this point starting at line 102.

Further research into the role of polyreactivity in anchor-binding mAbs is a future research interest and is currently outside the realm of this study. Notably, the distinction in polyreactivity across anchor antibodies allows us to dissect the factors that regulate polyreactivity in a genetically restricted class of mAbs.

7. It is quite intriguing to see the sharp contrast between anchor- and central stalk-targeting mAbs on ADCC activity (ED Fig. 2e and 2f). Can this be explained by the positioning of Fc and distance to the effector cells that some of the authors on this manuscript previously proposed (head vs. stalk)? It would be informative to have potential explanations or speculation on this point.

We agree with the reviewer and this is what we believe is happening. Related to point 5, we believe that the orientation of the antibody is likely restricting contact of the Fc with FcRs on effector cells. We have added this at line 95-97.

8. Since the anchor-targeting mAbs can recognize both pre-pandemic and pandemic H1 lineages it would be very interesting to retrospectively analyze the anchor-targeting B cell response when pH1N1 was introduced. The H-L Ig paired sequencing datasets in DeKosky et al. (Nat Biotechnol, 2013) may have some clues to address this question.

We thank the reviewer for this idea and have been interested by this question as well. To address this, we analyzed the repertoire of day 14 memory B cells from DeKosky et al. However, no B cells with the very specific repertoire features of anchor-binding mAbs we identified in our study were present within the DeKosky et al. dataset. Despite this, we were still able to address this question by quantifying the

proportion of anchor-binding mAbs induced by the 2009 pH1N1 monovalent vaccine (MIV) relative to repeat exposure cohorts. Both cohorts have previously been described in the publications below.

[10.1126/scitranslmed.aad0522](https://doi.org/10.1126/scitranslmed.aad0522); [10.1016/j.immuni.2020.10.005](https://doi.org/10.1016/j.immuni.2020.10.005); [10.1126/scitranslmed.abg4535](https://doi.org/10.1126/scitranslmed.abg4535)

As previously reported in these studies, first exposure to the pH1N1 was able to robustly induce antibodies against the HA stalk domain, with a larger proportion of isolated mAbs targeting the HA stalk domain from the 2009 MIV cohort relative to those individuals who have been repeatedly exposed (Extended Data Fig. 6c). Although not significant, more individuals from the 2009 MIV cohort had an isolated mAb against the anchor epitope relative to those in the repeat vaccine cohorts (Extended Data Fig. 6d). Therefore, these data suggest that in the absence of immunity against variable epitopes of the HA head, individuals can recall memory B cells against the anchor epitope. As the cHA vaccine was designed to remove the variable head epitopes, this immunogen may be primed to mimic the responses seen with the 2009 MIV.

Minor comments:

1. It is not clear to me how the authors determine the HA domain specificities (line 68 & ED Fig. 1a). All 358 mAbs are isolated by the same H1 HA probe?

We have added to the methods (located in the supplemental materials) how HA domain specificity was determined starting at line 56, which was determined if a mAb bound to a cH5/1 or cH6/1 antigen and lacked hemagglutination inhibition activity, a feature of mAbs targeting the head domain.

To clarify the second point, in the methods section under “monoclonal antibody production”, we summarize how B cells were isolated. Plasmablasts were not bait-sorted, as this population is already highly enriched for specificity against influenza viruses one week after vaccination or infection (Wrammert et al. Nature 2008). Only memory B cells were sorted using an antigen probe, either A/California/04/2009 H1 for 030-09M 1B06 or cH5/1 for B cells isolated at day 113 of the cHA vaccine trial.

2. The term “multiple donors” in lines 109-110 is misleading and should be replaced with “two donors”.

We have made this edit now at line 140.

3. Why did the authors give mAbs only 2 hours before the virus challenge in mice? There will not be enough time for IP-injected mAbs to be distributed in the airway and lung to prevent virus infection.

Giving antibody 2±1 hours prior to a lethal infection is standard within the field and is routinely used by our lab and others. Specific references are listed below. We do appreciate the reviewers point this treatment may not prevent the establishment of infection. Despite this, the term prophylactic is still proper, as it is given at a timepoint that should prevent disease.

<https://doi.org/10.1016/j.cell.2019.03.048>

<https://doi.org/10.1016/j.immuni.2020.08.015>

<https://doi.org/10.1016/j.cell.2018.03.030>

<https://doi.org/10.1016/j.chom.2016.05.014>

[10.1126/science.1205669](https://doi.org/10.1126/science.1205669)

4. Did the authors assess the escapability of the anchor epitope? It is probably possible to infer the fitness of the virus possessing mutations within the anchor epitope from existing DMS data.

This was an interesting point raised by the reviewer which led us on a deeper dive on the conservation of the anchor epitope. We assessed the conservation of these contact residues using deep mutational scanning based on previously published data in Doud and Bloom, *Viruses* 2016 (<https://doi.org/10.3390/v8060155>). The results of this showed that there was a lot of permissibility at these specific residues, suggesting that this epitope has high mutational tolerance. We have added these data to Extended Data Fig. 4s. Despite this, there is little evidence that viruses with mutations are selected for in nature. We performed a second analysis of H1 conservation at the 5 primary side-chain contacts using sequences from all known clades of H1 viruses, as described in Zhuang et al. ([10.1186/s12985-019-1188-7](https://doi.org/10.1186/s12985-019-1188-7)). Notably, this analysis showed that the five contacts were near universally conserved across H1 viruses, consistent with a lack of escape variants against the anchor epitope. We have replaced Fig. 2j with these data and moved the previous Fig. 2j data to Extended Data Fig. 4r. Summary of H1 viruses (n=47) analyzed are in Extended Data Table 5.

Referee #2 (Remarks to the Author):

A. Summary of key results

This study by Guthmiller et al. explored a distinct class of broadly-neutralizing antibodies that target the influenza hemagglutinin (HA) at the anchor epitope, a site at the base of the HA stalk near its membrane anchor. A panel of 358 human monoclonal antibodies (mAbs) was generated, and 50 mAbs deriving from 21 individuals target the anchor epitope. The anchor-epitope-targeting mAbs bind and neutralize diverse H1 influenza viruses from the past 40 years, as well as several mutant HAs with known central-stalk-epitope escape mutants. Electron microscopy structural studies confirm that the epitope is distinct from the central-stalk epitope, and mAbs make extensive contacts across the HA fusion peptide. Interestingly, the antibodies all bind in an “upward” angle, suggesting that the epitope may only be transiently exposed during HA flexing on the membrane, and possibly explaining the absence of ADCC observed by the anchor-epitope-targeting mAbs due to this angle of HA binding. Importantly, prophylactic studies in mice showed 100% protection by anchor-epitope-targeting mAbs from a lethal dose of H1N1 virus, similar to the protection seen by central stalk mAbs. Notably, sequence analyses of the anchor-epitope-targeting mAbs reveal highly restricted heavy and light chain repertoires, with one public clone identified across multiple participants. Moreover, of the 22

participants who received chimeric HA vaccines, which boosts HA stalk-directed antibodies, the large majority of participants generated B cells with anchor-binding antibody restricted repertoires. Electron microscopy polyclonal epitope mapping and competition ELISAs confirmed the presence and abundance of anchor-epitope-targeting antibodies in serum samples from these chimeric HA vaccine participants. Overall, this study uncovered the existence and protective potential of a public class of broadly neutralizing antibodies that target the conserved HA anchor epitope.

B. Originality and significance

This study by Guthmiller et al. substantially enhances our understanding and provides new context to two previous publications:

1. "Influenza hemagglutinin membrane anchor" by Benton et al. in PNAS in 2018.

DOI:10.1073/pnas.181092711

In this publication, the human monoclonal antibody FISW84 was shown via cryoEM studies to bind at an "upward" angle to a conserved epitope at the base of the HA stalk near the "membrane anchor."

FISW84 broadly neutralizes diverse H1 influenza viruses. While this study showed the first existence of the HA anchor epitope, it was unclear how rare anchor-epitope-targeting antibodies are in humans and whether this class of antibodies is protective in vivo. The current study by Guthmiller et al. revealed that these antibodies are common in humans and are protective in vivo.

2. "A chimeric hemagglutinin-based universal influenza virus vaccine approach induces broad and long-lasting immunity in a randomized, placebo-controlled phase I trial." By Nachbagauer et al. in Nat. Med. in 2021.

DOI:10.1038/s41591-020-1118-7

In this publication, volunteers received vaccinations with sequential chimeric HA vaccines, which redirects immune responses to the HA stalk region. Anti-stalk antibody levels were boosted to high titers. Passive transfer of vaccinated human serum antibodies into mice showed protection from weight loss induced by an H6 virus that would only be reactive to anti-stalk antibodies. While this study showed that chimeric HA vaccines could boost HA stalk-directed antibody levels, it was unclear whether these antibodies targeted the central stalk epitope or the anchor epitope. The current study by Guthmiller et al. revealed that chimeric HA vaccines do indeed induce antibodies against the anchor epitope in the majority of vaccinated individuals, and that anchor-epitope-targeting antibodies specifically are protective.

Thus, while the current study is not the first to describe the HA anchor epitope, it is the first to show that most people can develop antibodies against this epitope, the first to show that antibodies against this epitope are protective in vivo, and the first to show that chimeric HA vaccines induce antibodies against this epitope. Therefore, the current study provides the necessary knowledge and context to target this HA epitope in the development of universal flu vaccines.

C. Data and methodology

This study was a massive undertaking in experimentation. The data was robust and of high quality. There are some minor suggestions for improvements to data presentation (described below).

D. Appropriate use of statistics

Statistics appear appropriate.

E. Conclusions

The conclusions are well justified by the data presented.

F. Suggested improvements

- Lines 65 – 67 and Ext. Fig. 1a: This section is a bit misleading, as it states “vaccinated and naturally infected volunteers” but does not clarify that ~1/3 of volunteers were from a chimeric HA vaccine cohort, whose immune responses are intentionally directed by vaccination toward the HA stalk. Thus, saying that “nearly half” of the mAbs were directed at the HA stalk is misleading, as this would not be expected from a normal group of people who were naturally infected or received a seasonal flu vaccine. (Indeed, of the 52 anchor-epitope-targeting mAbs isolated in this study, 44 of them were derived from chimeric HA vaccine volunteers). It is recommended that the language be clarified, and to consider deleting the left half of Extended figure 1a, which is specific to this unique cohort.

We thank the reviewer for raising this concern and agree that it is a bit misleading in the 49% figure reported on the left-hand side of Extended Data Fig. 1a is skewed by several of the cohorts in our study. Instead of removing this all together, we have added the proportion of stalk+ mAbs per cohort to Extended Data Fig. 1b, which clearly show that certain cohorts, including the cHA vaccine cohort, had a higher proportion of mAbs against the stalk domain. We have modified the text starting at line 70 to clearly state this. In addition, we have added to line 88 that 34 of the anchor-binding mAbs were isolated from 15 donors within the cHA cohort.

- Line 152: “the anchor epitope is a common target of the human MBC repertoire against HA” should be clarified as “the anchor epitope is a common target of the human MBC repertoire following chimeric HA vaccination.”

We have made this revision to improve accuracy and is now located at line 188.

- Line 160: “the anchor epitope is common within the MBC pool and polyclonal serum antibody response” should be clarified as “the anchor epitope is common within the MBC pool and polyclonal serum antibody response in chimeric HA vaccinated individuals.”

To clarify, the serum antibody responses detected in serum were from two individuals within the 2014 QIV cohort, not the cHA cohort, that had generated a plasmablast response against the anchor epitope. We have edited this sentence, now starting at line 196, to read “...the anchor epitope is common within the MBC pool and polyclonal serum antibody response after vaccination.”

- Lines 162-174, Figure 4b,c, and Ext. Fig. 5c,d: the jargon should be simplified, as many people may not know what IIV, MIV, AS03, cH5/1, or cH8/1 mean, and it isn't really necessary to explain this. Instead, it is recommended to simplify to "adjuvanted cHA" to describe the results. Same with labels in Figure 4b,c, and Ext. Fig. 5c,d and their legends.

We feel that completely removing the context of what these vaccine formulations were, particularly within the cHA vaccine cohort, would substantially diminish the ability of readers to interpret results. To reduce confusion, we have added the vaccine group information to Figure 4a and Extended Data Fig. 6, defined each vaccine in the text and within a footnote associated with Extended Data Table 1. To improve clarity, we have also added a sentence starting at line 201 to define the different formulations of the cHA vaccine. In figure 4 and Extended Data Fig. 6, we changed the labels from the long confusing names to the vaccine groups.

- Figure 2e: lavender color is hard to see – it is suggested to change to another color

We have changed the color of the light chain contacts to yellow to make it easier to discern.

- Ext. Table 1: should include footnotes to explain what the different vaccines are.

As noted above, we have added these footnotes for Extended Data Table 1.

- Line 463: "in process of being deposited" – data should be deposited and made available by publication.

Samples have been deposited, including the 1952 sequences in Fig. 3a, and they will be publicly available by the time of publication.

G. References

- References 5 and 13 appear to be the same reference.

We thank the reviewer for pointing this out and we have fixed this error.

H. Clarity and context

Abstract, introduction, and conclusions are appropriate.

Referee #3 (Remarks to the Author):

This manuscript focuses on novel HA anchor epitope-binding antibodies. The authors first generated 358 mAbs from vaccinated and naturally infected volunteers and classified them according to their recognized epitopes. They identified 50 mAbs that bind to the anchor epitope of the HA stalk. They revealed that all of the identified anchor epitope-binding mAbs utilized unique repertoires that differ

from those of Central Stalk epitope-binding mAbs. In addition, the authors showed that anchor epitope-binding B cells are common among human memory B cells and a chimeric HA vaccine strategy recalls/activates these cells, suggesting that induction of these antibodies is key for protection against seasonal and pandemic influenza viruses. The manuscript is well written and the data are plentiful and well presented. However, the authors need to make it clearer that anchor epitope-binding antibodies are superior to previously identified broadly neutralizing antibodies. Also, some issues need to be addressed, as detailed below.

Major points:

1. Overall, the anchor epitope-binding antibodies are novel, but the binding affinities of these antibodies are less broad than those of Central Stalk epitope-binding mAbs.

The authors should show and discuss the superiority of anchor epitope-binding antibodies compared with previously identified Central Stalk epitope-binding mAbs.

First, we would like to thank the reviewer for their thoughtful and positive feedback on our manuscript. We do want to clarify though that it was not the purpose of our manuscript to suggest that the anchor epitope and antibodies against it are superior to that of antibodies against the central stalk epitope. Instead, our study highlights that the antibody response against the HA stalk is more complicated than previously reported and that anchor-targeting antibodies have been a missing puzzle piece of humoral immunity against the HA stalk. Importantly, the identification of an additional protective epitope is critical for universal influenza vaccine strategies, as our study highlights immunogen design and formulations that can promote their induction.

From a practical standpoint, the pan-H1 breadth of anchor-epitope targeting antibodies is important for the generation of protective humoral immunity against H1 viruses, which have been in circulation for 83 of the past 103 years and have led to 2 pandemics (and a 3rd H1N1 introduction in 1977 that was not classified as a pandemic) in this time frame. In addition, our data indicate that anchor antibodies can acquire H2-binding and neutralization, despite no opportunity to affinity mature against this antigen. These data suggest anchor-targeting B cells could be trained by vaccination to provide protection against H2N2 viruses, which is feared to reemerge and cause a pandemic. Lastly, evidence in the HIV-1 field has shown that immune focusing to a single epitope can lead to viral escape mutants in humans. Similarly, massive antibody responses against a single epitope of HA can promote the outgrowth of viruses that evade antibodies against that epitope (Linderman et al. PNAS 2014; Park et al. Nature Med. 2020). Therefore, the identification of additional broadly protective epitopes that can be targeted is an important endeavor. These points were added to the discussion starting at line 252.

2. Page 5, lines 98-100. Three anchor epitope-binding mAbs possess ADCC activity, as well as Central Stalk epitope-binding mAbs. What causes the different ADCC activity among the anchor epitope-binding mAbs?

This is an interesting point that is hard to resolve but we have several ideas. Notably, we believe that the upward angle of approach of anchor-binding mAbs limits the Fc portion from accessing FcRs on the surface of effector cells. In contrast, antibodies against the central stalk epitope and the group 2 stalk epitope (CR8020/CD8043) have ADCC activity and both approach their epitopes at a downward angle, which would make the Fc region more accessible. We have added to the text at line 95-97.

However, as far as why several anchor-binding mAbs have ADCC activity and others do not is less clear and could be a result of distinct VH/VK pairings. Two of the three antibodies with ADCC activity utilize VH3-48. However, analysis of the angle approach in Fig. 1b of 047-09 4F04 (VH3-48, ADCC positive) versus 241 IgA 2F04 (VH3-23, ADCC negative) showed there was no difference in the Fab binding angle, suggesting there was no likely reason the Fc would be more accessible by 047-09 4F04. We have not added this to the paper as it is too speculative. As we identify more anchor-binding mAbs, it will be of interest to understand how VH/VK pairing and angle of approach shape antibody effector functions.

3. Page 9, line198. The authors suggested that anti-anchor antibodies have the potential to cross-neutralize group 1 viruses including H2 and H5. However, as shown in Fig.1f, anti-anchor antibodies that can bind to H2 or H5 are quite rare. Although it is clear that a few of the anti-anchor antibodies cross-neutralize H2 or H5, the anchor epitope does not seem to be an important target for bnAbs against group 1 viruses. Therefore, the authors' discussion on this point is a little misleading.

We have added data to show that two of the H2-reacting mAbs can neutralize H2 (Extended Data Fig. 2c). Because of the genetic restriction and similar modes of binding to the fusion peptide (Fig 2 and Extended Data Fig. 4), our data suggest that upon H2 exposure, anchor-targeting B cells could become activated, affinity mature, and gain neutralizing potential against H2 viruses. This is mentioned at line 249.

4. Fig. 4. The authors should examine neutralizing titers against various H1 viruses. When serum neutralizes various H1 viruses, it is important to show how large the impact of anchor epitope-binding Abs is for the neutralization compared with Central Stalk epitope-binding Abs.

The data regarding serum neutralization breadth from the cHA vaccine trial were published in late 2020 (Nachbagauer et al. Nat. Med. 2020) and nicely showed (below; Extended Data Fig. 1) that both the kinetics and magnitude of the serum neutralizing antibody titers against a chimeric H6/1 virus, pH1N1, and an avian-swine H1N1 (asH1N1) virus matched the kinetics and magnitude of serum antibody titers against the anchor and CS epitopes (second image, Extended Data Fig. 6). Of note, the ch6/1N8 virus specifically measure neutralizing titers against the HA stalk domain and the avian-swine virus is very distantly related to pH1N1 viruses (Yang et al. Emerg. Infect. Dis. 2012; [10.3201/eid1807.120009](https://doi.org/10.3201/eid1807.120009)). Therefore, these data indicate the neutralizing titers largely target the stalk domain and appear to be tightly linked to titers against both the anchor and central stalk epitopes. We have added this point to the paper starting at line 208.

Nachbagauer et al. Nature Medicine, 2020 <https://doi.org/10.1038/s41591-020-1118-7>

Extended Data Fig. 1

Extended Data Fig. 6 – Kinetics of anti-anchor epitope (a) and anti-central stalk epitope (b) antibody responses across vaccine groups.

5. Extended Data. Fig. 2. The authors should examine the protective effect in terms of the therapeutic use of these anchor-epitope binding antibodies. Also, to investigate the potential of anchor epitope-targeting antibodies to provide protection against pan-H1 influenza virus, the authors should use multiple virus strains.

We have added to Extended Data Fig. 3b experiments where we therapeutically administered the antibody cocktails 48 hours after infection. At 5 mg/kg, 100% of mice survived a lethal infection with the pH1N1 virus, A/Netherlands/602/2009.

For studying protection against an additional H1 viruses, we prophylactically administered the antibody cocktails and infected mice with a lethal dose (10 LD50) of A/Fort Monmouth/1/1947 H1N1, a virus in circulation prior to the birth of any of the donors in our study. At 5 mg/kg, 70% of mice that received the anchor cocktail survived, relative to 90% of mice that received the central stalk cocktail. These data indicate that anchor-targeting antibodies have the potential to provide protection against lethal H1N1 virus infection.

Minor points:

1. Please provide a reference for CR9114 and FI6v3.

We have added these references.

2. Fig. 1c. It is hard to distinguish between the colors used for mapping.

We have changed the color of 241 IgA 2F04 to lavender color to contrast with the blue and teal colors used for the other two anchor antibodies. We think that it is now easier to evaluate the different footprints in Fig. 1c.

3. Fig. 1f. The authors should state how they defined mAbs binding.

We have added more detail on how antibodies were defined as targeting the stalk in the methods. We have also added at line 86 that an HA competition assay with 047-09 4F04 was used to identify mAbs that bound the anchor epitope.

4. Line 84. Please show the data, which would be informative.

We have added these data to Extended Data Fig. 2c.

5. Extended Data Fig. 1h and lines 90-91. The effect of the HA2 A44V mutant on the binding to the HA stalk is larger than that of the other stalk mutants. The authors should comment on this.

We completely agree that the effect of A44V results for both anchor and CS targeting antibodies were quite interesting. A recent paper by Park et al. (Nature Medicine 2020, link below), showed this mutation made structural changes to the epitope. As the anchor epitope is quite distant from this mutation, these conformational changes may in part negatively impact antibody binding. Despite this, the reduction in binding was marginal (10-30% reduction) and likely has little impact on neutralization by anchor-targeting mAbs. We have added this starting at line 113.

Noteworthy, we have changed the numbering for Extended Data Fig. 2e to be consistent with the numbering used in Fig. 2. A44V is now listed at A373V, which is based on the H3 Burke and Smith Numbering System ([10.1038/s41591-020-0937-x](https://doi.org/10.1038/s41591-020-0937-x)).

6. Extended Data. Fig. 2. To test whether mAbs targeting the anchor epitope were protective in vivo, the authors provided only the data of weight loss and survival of infected mice. However, viral titers in organs should also be tested to demonstrate the protective efficacy of the mAbs.

We determined lung viral titers at day 3 and day 6 post A/Netherlands/602/2009 H1N1 infection in mice that received 5 mg/kg of each treatment group prophylactically. Notably, there were no statistical differences in viral titers between mice that received the anti-anchor antibody cocktail and the negative control antibody. In addition, modest reductions in viral titers were noted for mice that received the central stalk cocktail.

Anti-stalk antibodies are typically neutralizing but typically do not provide sterilizing immunity, like antibodies targeting the HA head. As a result, antibodies against the stalk do not prevent infection altogether, but neutralize subsequent rounds and limit disease progression. Notably, a previous study by Sutton et al. ([10.1128/JVI.01603-17](https://doi.org/10.1128/JVI.01603-17)) found that there were limited differences in lung viral titers in mice treated with monoclonal antibodies against the central stalk relative to control treated mice. This was despite the fact the mice treated with the anti-stalk mAbs were protected from morbidity and mortality. Therefore, our data and those from Sutton et al. suggest that lung viral titers do not necessarily reflect disease, as these same treatment doses provided 100% protection against both morbidity and mortality.

7. Fig. 4 and Extended Data Fig. 5 a-d. The labels of the X-axis are a little confusing for readers. Please provide sufficient explanation of the abbreviations such as IIV and LAIV.

To reduce confusion, we have added the vaccine groups in Fig. 4 and Extended Data Fig. 6 and replaced the labels with the vaccine groups. We have also added these details in the main manuscript at lines 201 as well as have included a footnote for Extended Data Table 1.

8. Fig. 4e. please spell out “EC50”.

We have added the definition to Fig. 4 legend.

9. Fig. 5d, Y-axis. Characters are overlapping. Please correct.

This was Extended Data Fig. 5d and we have fixed this error. It is now Extended Data Fig. 6f.

10. Materials and Methods, page 2, lines 38-42. It is not appropriate for several authors to be acknowledged in this section.

We have removed mention of the A/Swine/Mexico/AVX8/2011 virus and the acknowledgements associated with it as well as mention of the Krammer, Coughlan, and Wilson laboratories.

Reviewer Reports on the First Revision:

Referee #1

The authors had adequately addressed all my previous concerns. Regarding the “upward” angle of approach I would like to suggest a few things that may add a little more depth in their discussion. In the FISW84 paper by Benton et al. (PNAS 2018), they observed a dramatic tilt of HA ectodomain relative to

its transmembrane domain as much as 52° in their cryoEM structure of HA with micelle. This may provide potential explanations on how the anchor epitope is accessed by antibodies (or B cells) in the context of membrane-anchored HA. Another interesting aspect of the “upward” angle of approach is that this may also be seen in other occasions. Indeed, one of the authors in the present manuscript reported the cryoEM structures of MERS S-2P with a S2 base-binding antibody G4 (Pallesen et al., PNAS 2017). This G4 antibody neutralizes MERS and approaches its epitope on the spike “upward” though the epitope is not as membrane proximal as the HA anchor epitope. Interestingly, coronavirus spike (SARS-CoV-2) was shown to tilt as much as 60° relative to its transmembrane domain just like HA (Ke et al., Nature 2020). Including discussion on these points and how the “upward” angle of approach would avoid crowded glycans and accessible-yet-hypervariable surface might further justify this type of antibodies.

Minor comments:

1. Line 62: “a previously unappreciated epitope” should be “an underappreciated epitope”. FISW84 defined the epitope.
2. Line 66: “Identification of anchor epitope” should be “Identification of anchor epitope-directed antibodies”. Again, the epitope was previously identified.
3. Line 100: “other viral subtypes” should be “other HA subtypes”.
4. Line 163: should “inter-Fab” be “intra-Fab”?
5. Fig. 4: vaccine group table in the panel a is confusing. I guess the authors meant the group 2 boost to be “IIV” instead of “IIV + AS03”.

Referee #2

he authors responded to my review appropriately for the most part; however, some issues remain, as detailed below:

To my comment, “viral titers in organs should also be tested to demonstrate the protective efficacy of the mAbs”, the authors responded, “Notably, there were no statistical differences in viral titers between mice that received the anti-anchor antibody cocktail and the negative control antibody.”

1. As it is interesting that the lung titers did not reflect morbidity and mortality, the authors should add more explanation in the text as they did in the response letter.
2. Extended Fig. 3c. Were these mice prophylactically treated with mAbs? The information is lacking in the figure legend. It would be informative to show the viral burden in the lungs of mice treated prophylactically as well as the mice treated therapeutically.
3. The authors analyzed Extended Fig. 3c by using two-tailed Mann-Whitney tests; it would be better to use a two-way ANOVA to analyze these data.
4. It would be better to include the statistical analysis with the survival data.

Author Rebuttals to First Revision:

Referee #1 (Remarks to the Author):

The authors had adequately addressed all my previous concerns. Regarding the “upward” angle of approach I would like to suggest a few things that may add a little more depth in their discussion. In the FISW84 paper by Benton et al. (PNAS 2018), they observed a dramatic tilt of HA ectodomain relative to its transmembrane domain as much as 52° in their cryoEM structure of HA with micelle. This may provide potential explanations on how the anchor epitope is accessed by antibodies (or B cells) in the context of membrane-anchored HA. Another interesting aspect of the “upward” angle of approach is that this may also be seen in other occasions. Indeed, one of the authors in the present

manuscript reported the cryoEM structures of MERS S-2P with a S2 base-binding antibody G4 (Pallesen et al., PNAS 2017). This G4 antibody neutralizes MERS and approaches its epitope on the spike “upward” though the epitope is not as membrane proximal as the HA anchor epitope. Interestingly, coronavirus spike (SARS-CoV-2) was shown to tilt as much as 60° relative to its transmembrane domain just like HA (Ke et al., Nature 2020). Including discussion on these points and how the “upward” angle of approach would avoid crowded glycans and accessible-yet-hypervariable surface might further justify this type of antibodies.

We appreciate the reviewer’s comment on this as these are interesting ideas. We have added a paragraph at line 288 to discuss these two points and why they may explain the restriction of mAbs targeting this epitope. Notably, the Benton et al. PNAS study identified a large N-glycan structure directly above the anchor epitope. This supports the notion that these mAbs need to bind the epitope from an upward angle to avoid conflict with this glycan. We have added the Pallesen et al. manuscript to the paper but due to length restrictions, did not discuss the Ke et al. Nature manuscript.

Minor comments:

1. Line 62: “a previously unappreciated epitope” should be “an underappreciated epitope”. FISW84 defined the epitope.

We have edited this line.

2. Line 66: “Identification of anchor epitope” should be “Identification of anchor epitope-directed antibodies”. Again, the epitope was previously identified.

We have changed this to “Discovery of anchor epitope binding mAbs”, as we are restricted to 40 character titles within the results section.

3. Line 100: “other viral subtypes” should be “other HA subtypes”.

We have edited this point.

4. Line 163: should “inter-Fab” be “intra-Fab”?

The reviewer is absolutely correct on this point and we have edited this.

5. Fig. 4: vaccine group table in the panel a is confusing. I guess the authors meant the group 2 boost to be “IIV” instead of “IIV + AS03”.

The reviewer is correct, group 2 is just the IIV. We have edited this error in the latest iteration.

Referee #2 (Remarks to the Author):

I have reviewed the revised version of this manuscript and feel that the points raised in my previous review have been satisfactorily addressed.

We sincerely thank the reviewer for their kind and thoughtful reviews.

Referee #3 (Remarks to the Author):

The authors responded to my review appropriately for the most part; however, some issues remain, as detailed below:

To my comment, “viral titers in organs should also be tested to demonstrate the protective efficacy of the mAbs”, the authors responded, “Notably, there were no statistical differences in viral titers between mice that received the anti-anchor antibody cocktail and the negative control antibody.”

1. As it is interesting that the lung titers did not reflect morbidity and mortality, the authors should add more explanation in the text as they did in the response letter.

We have added a more thorough explanation of why there are no differences starting at line 139. We kept this explanation more concise than the one given in the response letter, as we had limited space in the main text.

2. Extended Fig. 3c. Were these mice prophylactically treated with mAbs? The information is lacking in the figure legend. It would be informative to show the viral burden in the lungs of mice treated prophylactically as well as the mice treated therapeutically.

We apologize that this information was not in the figure legend and have added that the antibody was administered prophylactically to the figure legend now.

We did not observe significant reductions in viral lung titers in our prophylactic treatment model. Considering the short window between therapeutic treatment (48h) and relevant timepoints for viral lung titer quantification (72h/D3), we did not anticipate that we would observe any reduction in viral titer in the therapeutic treatment model. As such, samples were not harvested for analysis. We thank the reviewer for this suggestion, this is something we will consider for future work when we perform a more in-depth analysis of the precise correlates of protection for this new class of antibody.

3. The authors analyzed Extended Fig. 3c by using two-tailed Mann-Whitney tests; it would be better to use a two-way ANOVA to analyze these data.

We have corrected these statistics and are now showing the results of a Kruskal-Wallis test for each time point, which is the non-parametric version of a two-way ANOVA.

4. It would be better to include the statistical analysis with the survival data.
We have added these statistics to Extended Data Fig. 3a, b, and d.